# FGF4-FGFR1 signaling promotes podocyte survival and glomerular function to ameliorate diabetic kidney disease in male mice

Jie Zhou [1,2,6] ✉, Shuxin Wang[1,6], Jiaxin Lou[1,6], Beibin Pan[1], Min Zhao[1], Qian Li[1,2], Jing Zhou[1], Yali Du[1], Shuodan Ding[1], Meiling Yu[1], Jingjing Zhou[1], Xinwei Chen[1], Lingwei Jin[3], Xinyi Wang[4], Yepeng Hu[5], Zhe Wang[5], Xiaokun Li [1], Chao Zheng [5] ✉, Jian Sun [1] ✉ & Zhifeng Huang [1,2] ✉

Podocyte injury is central to diabetic kidney disease (DKD) pathogenesis, however, the mechanisms underlying podocyte loss remain unclear. Emerging evidence underscores the involvement of fibroblast growth factors (FGFs) in renal pathophysiology. Here we reveal a previously unappreciated role of podocyte-secreted FGF4 in safeguarding renal function. FGF4 expression is downregulated in renal tissues from DKD patients and animal models, correlating with disease severity. Podocyte-specific deletion of *Fgf4* exacerbated podocyte loss and accelerated DKD progression in mice. Conversely, treatment with recombinant FGF4 (rFGF4) improved glomerular filtration and reduced renal injury and fibrosis in diabetic male mice. These effects are primary mediated by activating the FGFR1-AMPK-FOXO1 signaling cascade in podocytes, which mitigates oxidative stress, suppresses apoptosis, and fosters podocyte survival. Notably, rFGF4 also restores the morphology and function of human podocytes exposed to high glucose. Our findings establish FGF4 as a critical regulator of podocyte homeostasis and a potential therapeutic target for DKD.

Diabetic kidney disease (DKD) represents a global health challenge, affecting approximately 30% of individuals with type 1 diabetes and 40% of those with type 2 diabetes, thereby becoming the leading cause of end-stage renal disease worldwide[1,2]. DKD is characterized by structural and functional kidney abnormalities, including glomerular hypertrophy, mesangial expansion, podocyte injury, and a gradual decline in renal function[3]. Current therapeutic strategies primarily focus on glycemic and lipid control, blood pressure management, and the use of angiotensin-converting enzyme inhibitors (ACEIs)[2,4,5]. Recent advancements have utilized novel glucose-lowering agents such as sodium-glucose cotransporter-2 (SGLT2) inhibitors, glucagon-like peptide-1 (GLP-1) receptor agonists, and non-steroidal mineralocorticoid receptor antagonists (ns-MRAs)[6,7]. Nevertheless, these interventions have limited efficacy in halting DKD progression,

[1]State Key Laboratory of Macromolecular Drugs and Large-scale Preparation, School of Pharmaceutical Sciences, Wenzhou Medical University, Wenzhou, Zhejiang 325035, China. [2]Translational Medicine Laboratory, The First Affiliated Hospital of Wenzhou Medical University, Wenzhou, Zhejiang 325035, China. [3]Department of Nephrology, The Second Affiliated Hospital, Yuying Children's Hospital of Wenzhou Medical University, Wenzhou, Zhejiang 325027, China. [4]Department of Endocrinology, Huangyan Hospital of Wenzhou Medical University, Taizhou, Zhejiang 325035, China. [5]Department of Endocrinology, The Second Affiliated Hospital, School of Medicine, Zhejiang University, Hangzhou, Zhejiang 310009, China. [6]These authors contributed equally: Jie Zhou, Shuxin Wang, Jiaxin Lou. ✉e-mail: zhoujie123@wmu.edu.cn; chao_zheng@zju.edu.cn; sunjian@wmu.edu.cn; hzf@wmu.edu.cn

underscoring the urgent need for novel therapeutic strategies that target the underlying mechanisms of kidney injury in diabetes.

The glomerulus plays a critical role in blood filtration, waste excretion, and fluid balance regulation. Podocytes, which are terminally differentiated epithelial cells within the glomerulus, are essential for maintaining the integrity of the glomerular filtration barrier[8]. In DKD, podocyte loss contributes to albuminuria, glomerulosclerosis, and progressive decline in renal function[9,10]. Podocyte detachment from the glomerular basement membrane and apoptosis are key factors driving podocyte loss and the progression of glomerular disease[11]. Despite their importance, effective approaches to prevent or reduce podocyte loss in DKD remain limited. Therefore, elucidating the mechanisms underlying podocyte injury is crucial to identify new therapeutic targets.

Recent studies have identified several endogenous protective factors expressed by podocytes, including adiponectin, pyruvate kinase M2 (PKM2), myeloid-derived growth factor (MYDGF), and sirtuin 6 (Sirt6), which are essential for maintaining podocyte function and glomerular filtration barrier integrity. Depletion of these factors exacerbates podocyte injury, detachment, and apoptosis, leading to increased albuminuria and more severe glomerular pathology[12-17]. In addition, the downstream signaling pathways and key nodes regulated by these endogenous factors play crucial roles in preserving podocyte function. Notably, AMP-activated protein kinase (AMPK) has emerged as a central player in this context[17-19]. As a vital cellular energy sensor, AMPK activation adaptively regulates glomerular volume, thereby preventing podocytopenia, especially in scenarios of podocyte injury[20]. This suggests that modulating AMPK activity could be instrumental in mitigating podocyte damage. The mammalian fibroblast growth factor (FGF) family comprises 18 structurally related polypeptides that play critical roles in development, tissue repair, and homeostasis[21-23]. Recent discoveries have underscored the involvement of FGFs in regulating glucose and lipid metabolism, positioning FGF signaling as a promising target for drug development in type 2 diabetes and related metabolic disorders[24-26]. Moreover, our previous findings indicate that the glucose-controlling and lipid-lowering effects of FGFs primarily depend on the AMPK signaling pathway in specific target cells such as muscle and liver parenchymal cells[27-29], prompting an in-depth investigation into the roles played by FGFs in the kidney, particularly in podocytes.

In the present study, we identify FGF4, a previously underinvestigated factor in renal function, as a critical regulator of podocyte integrity and glomerular filtration. Our findings demonstrate a progressive downregulation of FGF4 expression in podocytes as DKD advances. In diabetic mice, the absence of *Fgf4* in podocytes accelerates DKD progression. Conversely, pharmacological administration of recombinant FGF4 (rFGF4) alleviates oxidative stress and reduces podocyte apoptosis via the FGFR1-AMPK-FOXO1 signaling pathway. This intervention mitigates podocyte loss and consequently improves DKD. Notably, we extended these studies to clinical samples, including podocytes from DKD patients and isolated human renal glomeruli. In these human models, rFGF4 effectively reverses morphological abnormalities, reduces oxidative stress, and prevents podocyte loss under high glucose conditions, thus providing robust evidence that FGF4 is a promising therapeutic target for DKD.

## Results

### Inverse correlation of renal FGF4 expression with DKD progression

To elucidate the roles of FGFs in DKD, the expression profiles of all FGF family members were evaluated in renal tissues from both streptozotocin (STZ)-induced type 1 diabetic (T1D) mice and *db/db* type 2 diabetic (T2D) mouse models. Our analysis identified significant downregulation of *Fgf1* and *Fgf4* mRNA expression in renal tissues of diabetic mice compared to non-diabetic controls, with *Fgf4* exhibiting a

more pronounced reduction (Fig. 1a, b). RNA-Scope in situ hybridization confirmed the substantial decrease in renal *Fgf4* mRNA levels in both DKD models, and revealed that *Fgf4* mRNA localized predominantly to podocytes within the glomerular region (Fig. 1c, d and Supplementary Fig. 1a-d). Relatively low expression levels were observed in tubular epithelial cells (Supplementary Fig. 1a). Additionally, immunoblot analysis demonstrated a significant reduction in FGF4 protein levels in diabetic kidneys relative to non-diabetic controls (Fig. 1e, f).

We further examined renal tissues from DKD patients, using adjacent normal renal tissues from patients who underwent partial nephrectomy as controls (detail information available in Supplementary Table 1). In line with our observations in diabetic mice, there was a significant reduction in *FGF4* expression in renal tissues of DKD patients, and *FGF4* expression was predominantly localized to podocytes (Fig. 1g and Supplementary Fig. 1e). To explore the clinical relevance of these findings, primary podocytes were isolated from the urine of diabetic patients, including those with and without DKD. Immunofluorescence analysis revealed a substantial decrease in FGF4 protein levels in podocytes from DKD patients compared to those without DKD (Fig. 1h). Further, we performed stratified correlation analysis between urinary albumin-to-creatinine ratio (UACR) and podocyte FGF4 immunofluorescence intensity across stages of DKD progression (Phase II-IV). This revealed a significant stage-dependent correlation (Fig. 1i), suggesting that lower FGF4 expression is associated with severe renal injury and podocyte dysfunction in DKD.

To understand the transcriptional mechanisms governing FGF4 expression in DKD, we integrated predictions from bioinformatic databases, including GTRD, Cistrome, JASPAR, and Animal TFDB. Transcription factor CP2-like 1 (Tfcp2l1) was identified as a potential regulator of FGF4 expression (Supplementary Fig. 2a). qRT-PCR results showed that renal *Tfcp2l1* mRNA levels were significantly decreased in both *db/db* and STZ-induced diabetic mice (Supplementary Fig. 2b). To validate the role of Tfcp2l1 in regulating FGF4 levels, MPC-5 cells were transfected with *Tfcp2l1* small-interfering RNA (si*Tfcp2l1*) or a negative control siRNA. Knockdown of *Tfcp2l1* significantly reduced *Fgf4* mRNA and protein levels (Supplementary Fig. 2c, d). In addition, a wild-type (WT) Tfcp2l1 element in the *Fgf4* promoter exhibited strong responsiveness to Tfcp2l1 overexpression, driving expression of luciferase reporter (Supplementary Fig. 2e). Collectively, these results suggest that FGF4 is closely associated with DKD and plays a crucial role in its progression.

### Podocyte-specific deletion of *Fgf4* exacerbates renal injury under diabetic conditions

To understand the roles of FGF4 in the glomeruli and how its downregulation contributes to DKD pathogenesis, we generated a mouse model with podocyte-specific knockout of *Fgf4* (*Fgf4*-PKO) (Fig. 2a, b and Supplementary Fig. 3a). Blood glucose levels remained comparable between *Fgf4*-PKO mice and WT mice under both non-diabetic (-STZ) and diabetic (+STZ) conditions (Fig. 2c). However, biochemical markers of renal function, such as the UACR and blood urea nitrogen (BUN), were significantly elevated in *Fgf4*-PKO mice compared to WT littermates under diabetic conditions (Fig. 2d). The glomerular filtration rate (GFR), a more sensitive indicator of kidney function deterioration, was significantly reduced in WT diabetic mice compared to non-diabetic controls, as indicated by the prolonged metabolic half-life ($t_{1/2}$) of a fluorescent dye. Compared to WT diabetic controls, the GFR was further diminished by ~32% and the $t_{1/2}$ was extended by ~38% in *Fgf4*-PKO diabetic mice. These results suggest that *Fgf4* deletion exacerbates the impairment of renal filtration under diabetic challenge (Fig. 2e).

Histological analyses, including hematoxylin and eosin (H&E), periodic acid-schiff (PAS), and Masson's trichrome staining, revealed minimal impact of podocyte *Fgf4* deficiency on renal histology under

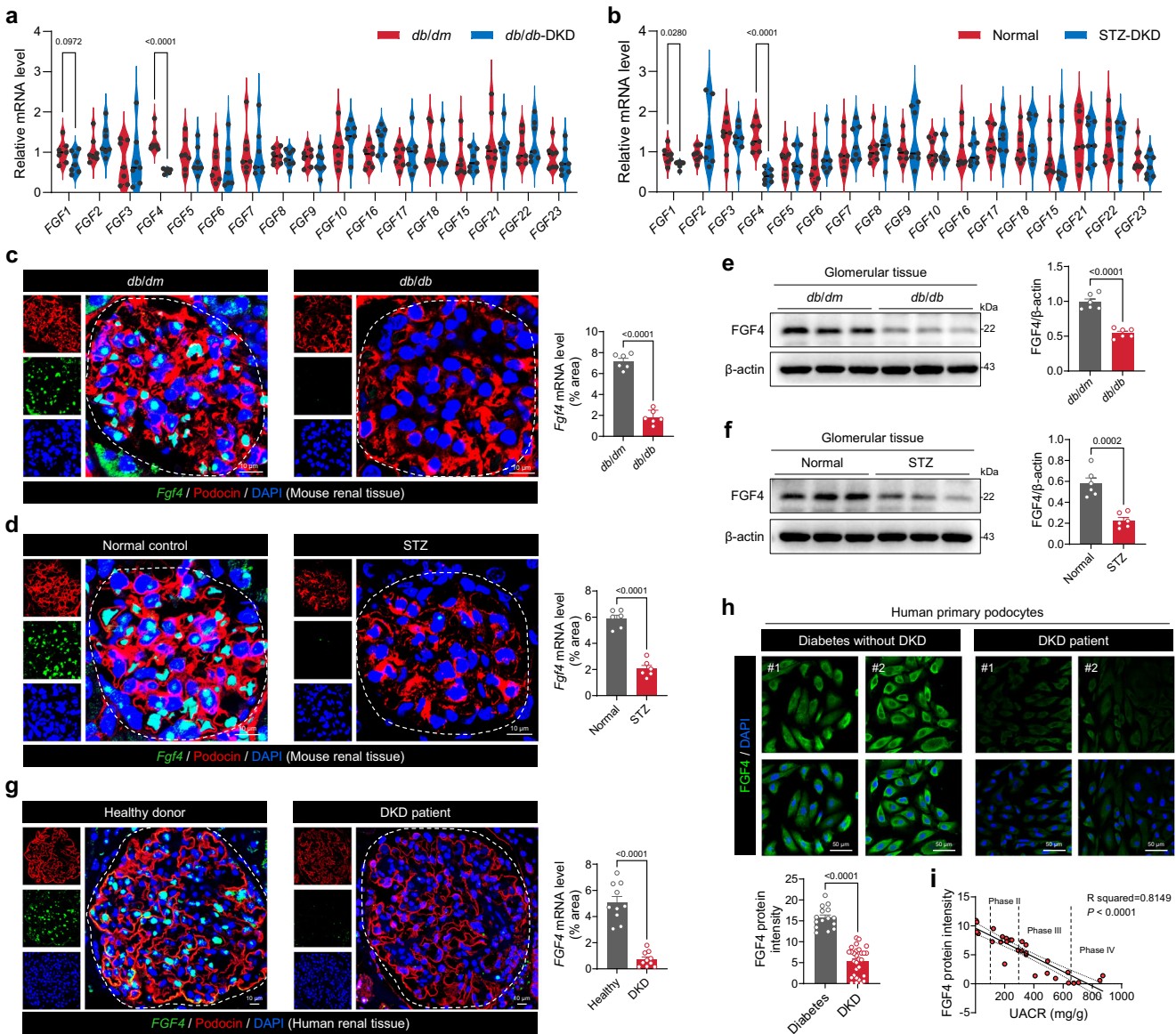

**Fig. 1 | Inverse correlation between renal FGF4 expression and DKD progression. a, b** Relative mRNA levels of all FGF family members in renal tissues from *db/db* mice (**a**) and STZ-induced diabetic mice (**b**) (n = 7). **c, d** RNA-Scope analysis of *Fgf4* mRNA expression (green) in glomerular podocytes from (**c**) *db/db* mice and (**d**) STZ-induced diabetic mice (n = 6). Samples are counterstained for the podocyte-specific marker podocin (red). **e, f** Western blot analysis of FGF4 protein levels in isolated glomerular lysates from the indicated groups (n = 6). β-actin served as a loading control. **g** RNA-Scope analysis of *FGF4* mRNA expression (green) in human renal tissues from healthy donors and DKD patients (n = 10). **h** Immunofluorescence analysis of FGF4 protein levels (green) in primary human podocytes isolated from diabetic patients with (n = 29) and without (n = 15) nephropathy. **i** Correlation analysis between UACR and FGF4 immunofluorescence intensity in DKD patients (n = 29). Nuclei were counterstained with DAPI (blue). Data are presented as mean ± s.e.m. *p < 0.05, ***p < 0.001, ****p < 0.0001 as determined by a two-sided unpaired Student's t-test (**a**, **b**) or unpaired two-tailed Student's t-test (**c**–**h**). Data in panel (**i**) were analyzed by two-sided simple linear regression with 95% confidence intervals calculated using the t-distribution, and no adjustment for multiple comparisons. UACR, Urinary albumin-to-creatinine ratio.

normal conditions. Conversely, under diabetic conditions, the deletion of podocyte *Fgf4* resulted in a more prominent expansion of the glomerular mesangium, glycogen deposition, and collagen accumulation (Fig. 2f). Transmission electron microscopy (TEM) further revealed thickening of the glomerular basement membrane (GBM), foot process fusion, and even foot process disappearance in both WT and *Fgf4*-PKO diabetic mice, with the latter exhibiting more severe pathological alterations (Fig. 2g). The expression levels of Nephrin and Podocin, integral components of the podocyte slit diaphragm[30,31], were significantly downregulated in diabetic mice (compared to non-diabetic controls), with lower levels seen in *Fgf4*-PKO diabetic mice than in WT diabetic littermates (Fig. 2h). To further investigate podocyte injury, we isolated glomeruli from all experimental groups (Supplementary Fig. 3b) and found that podocyte-specific *Fgf4* deficiency led to increased podocyte apoptosis, the accumulation of reactive oxygen species (ROS), and greater loss of podocytes in DKD mice (Fig. 2i).

To determine whether *Fgf4* deficiency affects podocyte function in a cell autonomous manner, siRNA was employed to silence *Fgf4* expression in murine podocytes cultured under high-glucose conditions (Supplementary Fig. 4a). Knockdown of *Fgf4* increased apoptosis and oxidative stress in high-glucose conditions (Supplementary Fig. 4b). Consistent with these observations, the absence of *Fgf4* exacerbated high glucose-induced upregulation of pro-

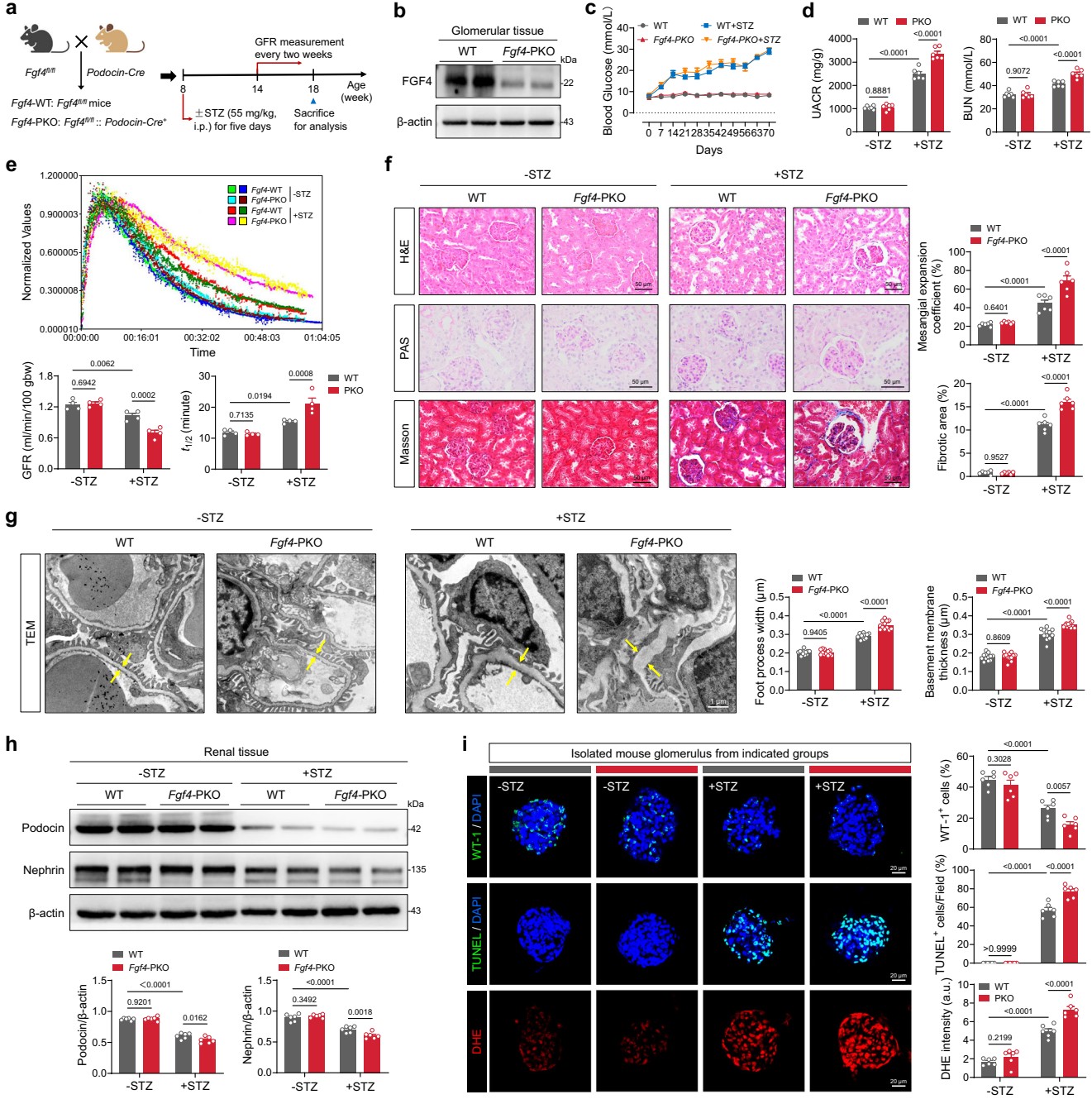

**Fig. 2 | Podocyte-specific deletion of *Fgf4* exacerbates diabetes-induced renal injury. a** Schematic of podocyte-specific knockout of *Fgf4* (*Fgf4*-PKO). At 8 weeks of age, male mice received an intraperitoneal (i.p.) injection of STZ (55 mg/kg) for five consecutive days. Blood glucose levels were measured weekly for 6 weeks. **b** Western blot analysis of FGF4 expression in glomeruli isolated from *Fgf4*-PKO and floxed control (WT) mice. **c** Blood glucose levels in WT and *Fgf4*-PKO mice (*n* = 8). **d** Alterations in UACR and BUN in renal tissues of WT and *Fgf4*-PKO mice (*n* = 6). **e** Renal excretion kinetics measured using fluorescein isothiocyanate–sinistrin, with GFR and metabolic half-life ($t_{1/2}$) quantified in each group (*n* = 4). **f** Representative images and quantification analysis of renal tissue sections stained with H&E, PAS, and Masson's trichrome (*n* = 6). **g** TEM and quantification of renal

tissues from the indicated groups (*n* = 12). Yellow arrows refer to basement membranes. **h** Western blot and quantification of Nephrin and Podocin levels in total renal tissue lysates from the indicated groups (*n* = 6). β-actin served as a loading control. **i** Representative images of isolated mouse glomerulus stained with DHE (red), as well as antibodies against Nrf-2 (pink) and WT-1 (green) (*n* = 6). Nuclei were counterstained with DAPI (blue). Data are presented as mean ± s.e.m. *$p < 0.05$, **$p < 0.01$, ***$p < 0.001$, ****$p < 0.0001$ as determined by ordinary two-way ANOVA followed by Sidak's multiple comparisons tests (**d**–**i**); ns, not significant. BUN, Blood urea nitrogen; GFR, Glomerular filtration rate; H&E, Hematoxylin and eosin; PAS, Periodic acid-Schiff; TEM, Transmission electron microscopy; WT-1, Wilms' Tumor 1.

apoptotic proteins and downregulation of antioxidant stress-related proteins in podocytes (Supplementary Fig. 4c). Taken together, these data from mouse and cellular models demonstrate that podocyte-specific *Fgf4* deficiency exacerbates podocyte-associated renal injury in DKD.

## rFGF4 administration ameliorates DKD pathologies

To gain insights into the pathophysiological role of FGF4 in renal tissues, structurally modified, non-mitogenic recombinant human FGF4 (rFGF4) was intraperitoneally (i.p.) administered to *db/db* mice at 1.0 mg/kg body weight every other day for 8 weeks (Fig. 3a). Consistent

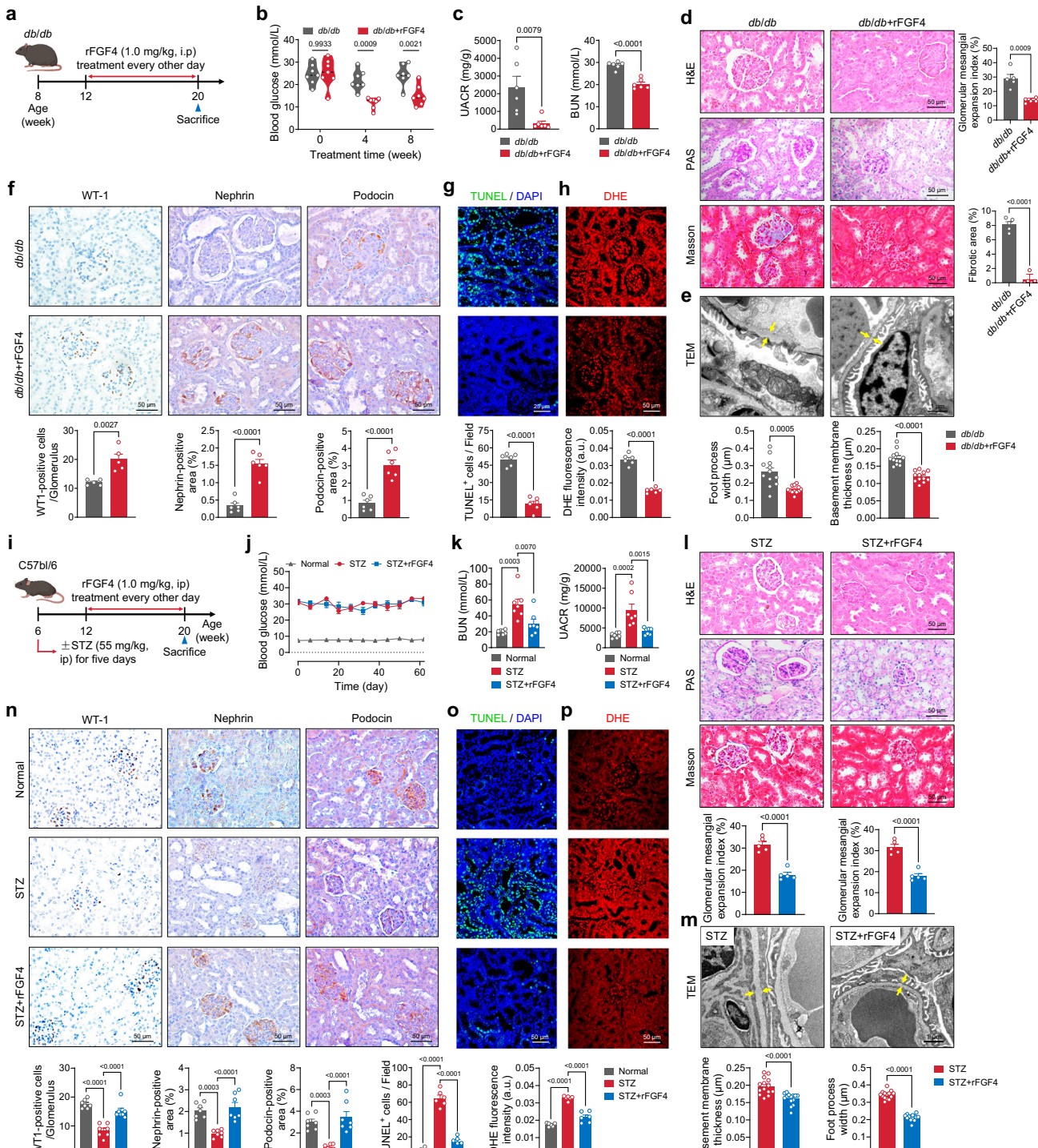

**Fig. 3 | rFGF4 alleviates glomerular injury in DKD by inhibiting oxidative stress and apoptosis in podocytes. a–h** *db/db* mice were treated with rFGF4 or PBS vehicle. **a** Schematic of rFGF4 treatment regimen: Male *db/db* mice at 12 weeks of age received intraperitoneal injections of rFGF4 (1.0 mg/kg) or vehicle every other day for 8 weeks. **b, c** Quantitative assessments of (**b**) blood glucose levels and (**c**) UACR and BUN (*n* = 6). **d** Representative renal sections stained with H&E, PAS, and Masson's trichrome (*n* = 5). **e** TEM analysis of renal tissues, with yellow arrows indicating the GBM (*n* = 12). **f** Immunofluorescence staining and quantification of podocyte markers WT-1, Nephrin, and Podocin across treatment groups (*n* = 5-6 per group). **g, h** Representative images and quantification of TUNEL (green) and DHE (red) staining, with nuclei counterstained with DAPI (blue) (*n* = 6). **i–p** STZ-induced diabetic mice were treated with rFGF4 or vehicle control. **i** Schematic of rFGF4 treatment regimen: Male mice at 6 weeks of age received i.p. injections of STZ

(55 mg/kg body weight) for five consecutive days, followed by treatment with rFGF4 (1.0 mg/kg body weight) or vehicle control every other day for 8 weeks starting at 12 weeks of age. Non-diabetic mice served as normal controls. **j, k** Measurements of blood glucose levels (**j**, *n* = 10), UACR, and BUN (**k**, *n* = 7). **l** Representative images of renal sections stained with H&E, PAS, and Masson's trichrome (*n* = 5). **m** TEM analysis of renal tissues, with yellow arrows indicating the GBM (*n* = 14). **n** Immunohistochemical analysis of podocyte markers in renal sections (*n* = 7). **o, p** Representative images and quantification of TUNEL (green) and DHE (red) staining in renal sections (*n* = 6). Data are presented as mean ± s.e.m. *p < 0.05, **p < 0.01, ***p < 0.001, ****p < 0.0001 as determined by ordinary two-way ANOVA followed by Sidak's multiple comparisons test (**b**), unpaired two-tailed Student's *t*-test (**c–h, l, m**), or ordinary one-way ANOVA followed by Dunnett's multiple comparisons test (**j, k, n–p**). GBM, Glomerular basement membranes.

with our previous findings[26, 28], rFGF4 significantly reduced blood glucose levels in *db/db* mice (Fig. 3b). Of note, rFGF4 treatment substantially decreased levels of the UACR and BUN in *db/db* mice (Fig. 3c). Histological assessments revealed that rFGF4 effectively mitigated renal mesangial expansion, glycogen accumulation, and fibrosis in *db/db* mice (Fig. 3d). Electron microscopy further revealed that rFGF4 treatment markedly prevented diabetes-induced glomerular injury, characterized by disruption of podocyte foot processes and basement membrane thickening (Fig. 3e).

Podocyte loss has been recognized as a critical driver of DKD progression[32–35]. Importantly, rFGF4 administration to *db/db* mice led to a significant upregulation of renal podocyte markers, including Wilms' Tumor 1 (WT-1), Nephrin, and Podocin, suggesting a reduction in podocyte loss and dysfunction (Fig. 3f). A dramatic reduction in ROS and apoptotic cells was observed in rFGF4-treated renal sections, as demonstrated by dihydroethidium (DHE) and terminal deoxynucleotidyl transferase-mediated dUTP-biotin nick-end labeling (TUNEL) staining (Fig. 3g, h). This was corroborated by decreased renal superoxide dismutase (SOD) activity and malondialdehyde (MDA) content (Supplementary Fig. 5a). rFGF4 treatment also attenuated the expression of pro-apoptotic factors cleaved-Caspase3 (c-Casp3) and Bax, while enhancing levels of the antioxidant proteins Nrf-2 and HO-1 in renal tissues (Supplementary Fig. 5b). Additionally, rFGF4 treatment reduced renal fibrosis, as evidenced by decreased protein levels of the profibrotic markers type IV collagen (Col-IV) and transforming growth factor-β (TGF-β) (Supplementary Fig. 5c, d), as well as reduced mRNA levels of *TGF-β*, *Col-IV*, and *α-SMA* (Supplementary Fig. 5e).

Next, we treated STZ-induced T1D mice with rFGF4 (1.0 mg/kg body weight) every other day for 8 weeks and measured the effects on DKD (Fig. 3i). Blood glucose levels were significantly increased in mice following STZ induction, confirming establishment of the T1D model. However, rFGF4 treatment failed to reduce blood glucose level in this T1D model (Fig. 3j). As observed in *db/db* mice, rFGF4 ameliorated STZ-induced elevations in UACR and BUN (Fig. 3k), as well as the histopathological features of kidney injury (Fig. 3l). Notably, administration of rFGF4 significantly ameliorated STZ-induced glomerular damage and podocyte loss in diabetic glomeruli (Fig. 3m, n). Further, rFGF4 treatment resulted in reduced levels of apoptosis and oxidative stress in renal tissues of STZ-induced diabetic mice (Fig. 3o, p). This was accompanied by the downregulation of apoptosis markers, as well as the upregulation of antioxidant defense proteins (Supplementary Fig. 5f, g). Concomitantly, a significant reversal of renal fibrosis was observed (Supplementary Fig. 5h-j). These results indicated that rFGF4 may ameliorate the pathologies of DKD via a mechanism independent of its insulin-sensitizing effects.

To evaluate the long-term effects and safety profile of rFGF4, we performed comprehensive assessments in both DKD models. Our results revealed that serum tumor markers (AFP, PSA, CA125, and CEA) remained within normal ranges without treatment-related aberrations (Supplementary Fig. 6a, e). No detectable increases in Ki67-positive foci or cell cycle gene expression were observed (Supplementary Fig. 6b, c, f, g). Histopathological evaluation of major organs (including heart, liver, spleen, pancreas, and lungs) using H&E staining revealed significant attenuation of hepatic lipid accumulation in rFGF4-treated *db/db* mice, consistent with our previous findings[27], However, no pathological alterations were observed in other major organs compared to untreated diabetic controls (Supplementary Fig. 6d, h). Further, pharmacokinetic studies demonstrated that intraperitoneal administration of rFGF4 achieved effective enrichment in both systemic circulation and renal tissues (Supplementary Fig. 7). Collectively, these findings establish that rFGF4 attenuates podocyte detachment and glomerular injury by suppressing oxidative stress and apoptosis, with a favorable safety profile.

## rFGF4 targets podocyte FGFR1 to protect renal function in diabetic conditions

Among the four principal FGF receptors, FGF receptor 1 (FGFR1) was the most abundantly expressed isoform in mouse renal tissues (Fig. 4a). Subsequent analysis demonstrated that FGFR1 was mainly located in the glomerular region in normal C57BL/6 mice (Fig. 4b and Supplementary Fig. 8a, b), a pattern also observed in human kidney tissues (Fig. 4c). Colocalization analyses further revealed that FGFR1 was predominantly expressed in glomerular podocytes in both normal and DKD samples, as evidenced by significant overlap with podocin, a podocyte marker (Fig. 4d, e). Minimal overlap was seen with the mesangial cell marker α-SMA and the endothelial cell marker CD31 (Supplementary Fig. 8c, d).

To ascertain whether rFGF4 protects against renal injury through podocyte FGFR1, podocyte-specific *Fgfr1* knockout (*Fgfr1*-PKO) mice were generated and subjected to STZ-induced diabetes, followed by rFGF4 or PBS vehicle treatment (Fig. 4f and Supplementary Fig. 9a, b). While STZ-induced diabetes resulted in elevated blood glucose levels, no significant differences were observed between *Fgfr1*-PKO and WT control mice (Fig. 4g). Body weight was likewise unaffected by *Fgfr1* deletion (Supplementary Fig. 9c). Treating diabetic WT mice with rFGF4 decreased the BUN and UACR, and increased the GFR; however, these effects were abolished in *Fgfr1*-PKO diabetic mice (Fig. 4h, i). Histological analysis revealed that *Fgfr1* deletion negated the protective effects of rFGF4, including the reduction of mesangial matrix expansion, glycogen deposition, and collagen accumulation in STZ-induced diabetic mice (Fig. 4j, k). Electron microscopy confirmed that rFGF4 improved podocyte structural integrity in diabetic mice, but this effect was nullified in *Fgfr1*-PKO mice (Supplementary Fig. 9d). Additionally, rFGF4-induced upregulation of Nephrin did not occur in *Fgfr1*-PKO diabetic mice (Fig. 4l, m). Glomeruli isolated from rFGF4-treated diabetic mice further demonstrated that rFGF4-induced upregulation of WT-1 and Nrf-2, as well as ROS reduction, were abrogated in *Fgfr1*-PKO mice (Fig. 4n, o). Moreover, rFGF4 failed to upregulate antioxidative stress factors in glomeruli of *Fgfr1*-PKO mice (Supplementary Fig. 9e).

Consistent with our in vivo findings, siRNA-mediated silencing of *Fgfr1* in MPC-5 cells disrupted the ability of rFGF4 to mitigate apoptosis and reduce ROS accumulation (Supplementary Fig. 10a-c). Further, *Fgfr1* deficiency not only abrogated rFGF4-induced elevation of anti-apoptotic and antioxidative stress markers but also exacerbated pro-apoptotic signals (Supplementary Fig. 10d, e). Collectively, these in vivo and in vitro data conclusively demonstrate that rFGF4 protects against oxidative stress and apoptosis of podocytes in DKD by activating FGFR1.

## Activation of the AMPK-FOXO1 pathway underlies the protective effects of FGF4 in DKD

To unravel the potential mediator downstream of the FGF4-activated FGFR1, RNA-Seq analysis was performed on renal tissue from STZ-induced diabetic mice treated with rFGF4 or PBS vehicle. In-depth analysis revealed 190 upregulated genes and 134 downregulated genes (Fig. 5a). KEGG Pathway enrichment analyses indicated that the AMPK signaling pathway was one of the most prominently upregulated pathways associated with apoptosis and oxidative stress (Fig. 5b). Additionally, we observed a pronounced enrichment and upregulation of forkhead box protein O1 (FOXO1), a downstream target of the AMPK pathway, in renal tissues of rFGF4-treated mice (Fig. 5c-d). In glomeruli isolated from STZ-induced diabetic mice, phosphorylation levels of AMPK and FOXO1 were significantly reduced compared to levels seen in non-diabetic controls; this reduction was more pronounced in *Fgf4*-PKO diabetic mice (Fig. 5e). Conversely, rFGF4 treatment significantly increased levels of phosphorylated (p)-AMPK and p-FOXO1 in renal tissues of STZ-induced diabetic mice (Fig. 5f). Moreover, rFGF4-induced activation of the AMPK-FOXO1 pathway was abolished in the

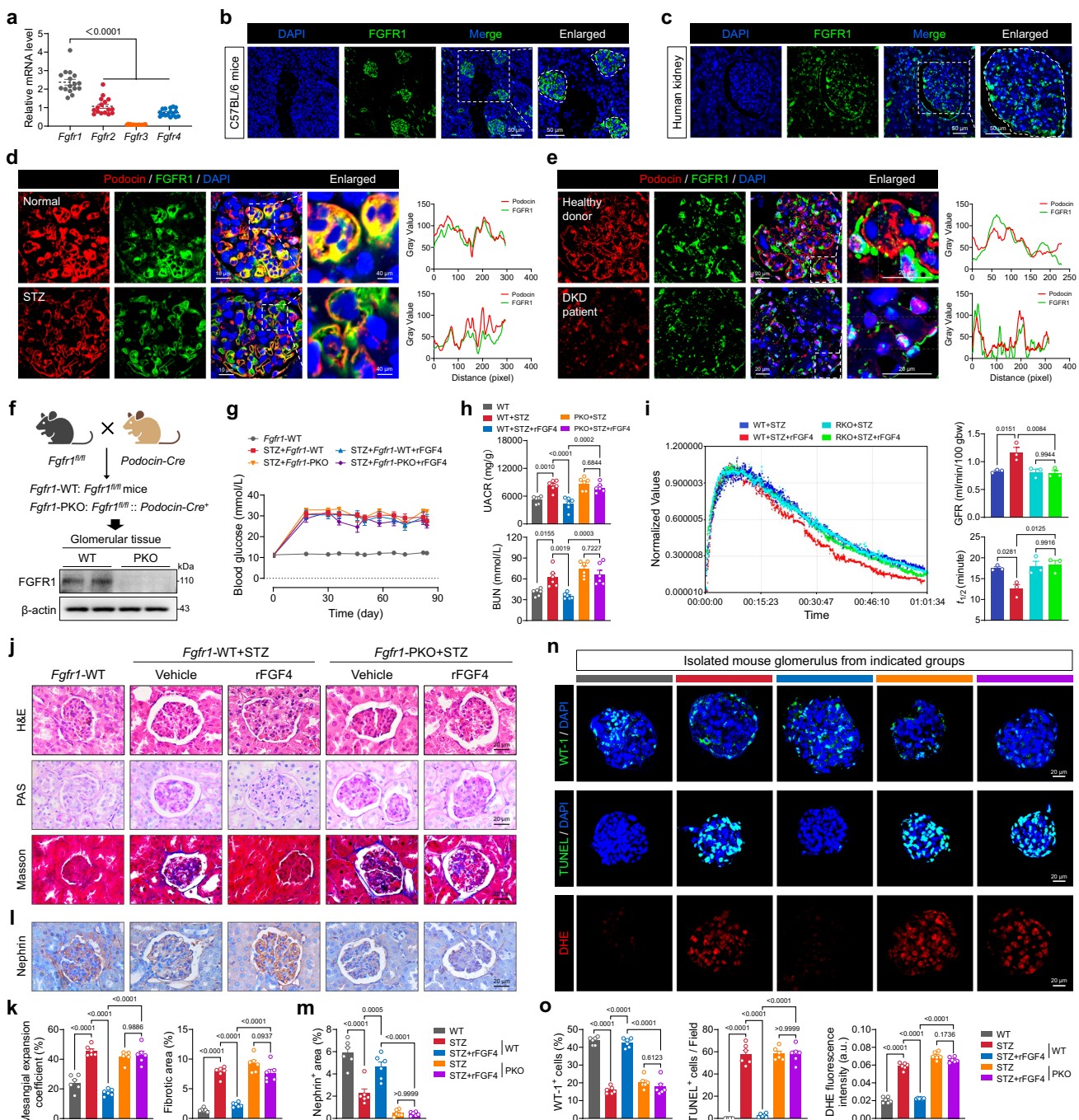

**Fig. 4 | FGFR1 mediates the therapeutic effects of rFGF4 in podocytes. a** Relative mRNA levels of *Fgfr1-4* in total renal tissue lysates from C57BL/6 mice (*n* = 16). **b**, **c** Immunofluorescence images depicting FGFR1 distribution (green) in normal renal tissues from (**b**) C57BL/6 mice and (**c**) human samples. White dashed lines delineate glomerular boundaries. **d**, **e** High-magnification images showing co-localization of FGFR1 (green) with the podocyte marker podocin (red) in (**d**) mouse and (**e**) human glomeruli. **f** Schematic of podocyte-specific *Fgfr1* knockout (*Fgfr1*-PKO), and Western blot of isolated glomeruli confirms loss of FGFR1 in *Fgfr1*-PKO versus littermate controls. **g**, **h** Changes in (**g**) blood glucose levels and (**h**) BUN and UACR across treatment groups (*n* = 6). **i** Renal excretion kinetics, GFR, and elimination half-life ($t_{1/2}$) in mice from each group (*n* = 3). **j**, **k** Histological examination and quantification of renal sections using H&E, PAS, and Masson's trichrome staining (*n* = 6). **l**, **m** immunohistochemistry and quantification of renal sections for the podocyte marker Nephrin from the indicated groups (*n* = 6). **n**, **o** Representative images of isolated mouse glomeruli stained with WT-1 (green), TUNEL (green), and DHE (red), with quantification shown in (**o**) (*n* = 6). Nuclei were counterstained with DAPI (blue). Data are presented as mean ± s.e.m. *$p < 0.05$, **$p < 0.01$, ***$p < 0.001$, ****$p < 0.0001$ as determined by ordinary one-way ANOVA followed by Tukey's multiple comparisons test (**a**–**o**); ns, not significant.

glomeruli of *Fgfr1*-PKO diabetic mice (Fig. 5g). Given that the FOXO1 transcription factor requires translocation into the nucleus to activate target genes[36], fractionation experiments were conducted to assess the influence of rFGF4 on FOXO1 subcellular distribution. We found that rFGF4 promoted the nuclear localization of FOXO1 in MPC-5 cells (Fig. 5h). Similarly, immunofluorescence assays confirmed the nuclear localization of FOXO1 after rFGF4 treatment, whereas FOXO1 was found in the cytosol in *Ampk*-silenced MPC-5 cells (Fig. 5i). These data suggest that rFGF4 modulates FOXO1 nuclear localization through AMPK activation in an FGFR1-dependent manner.

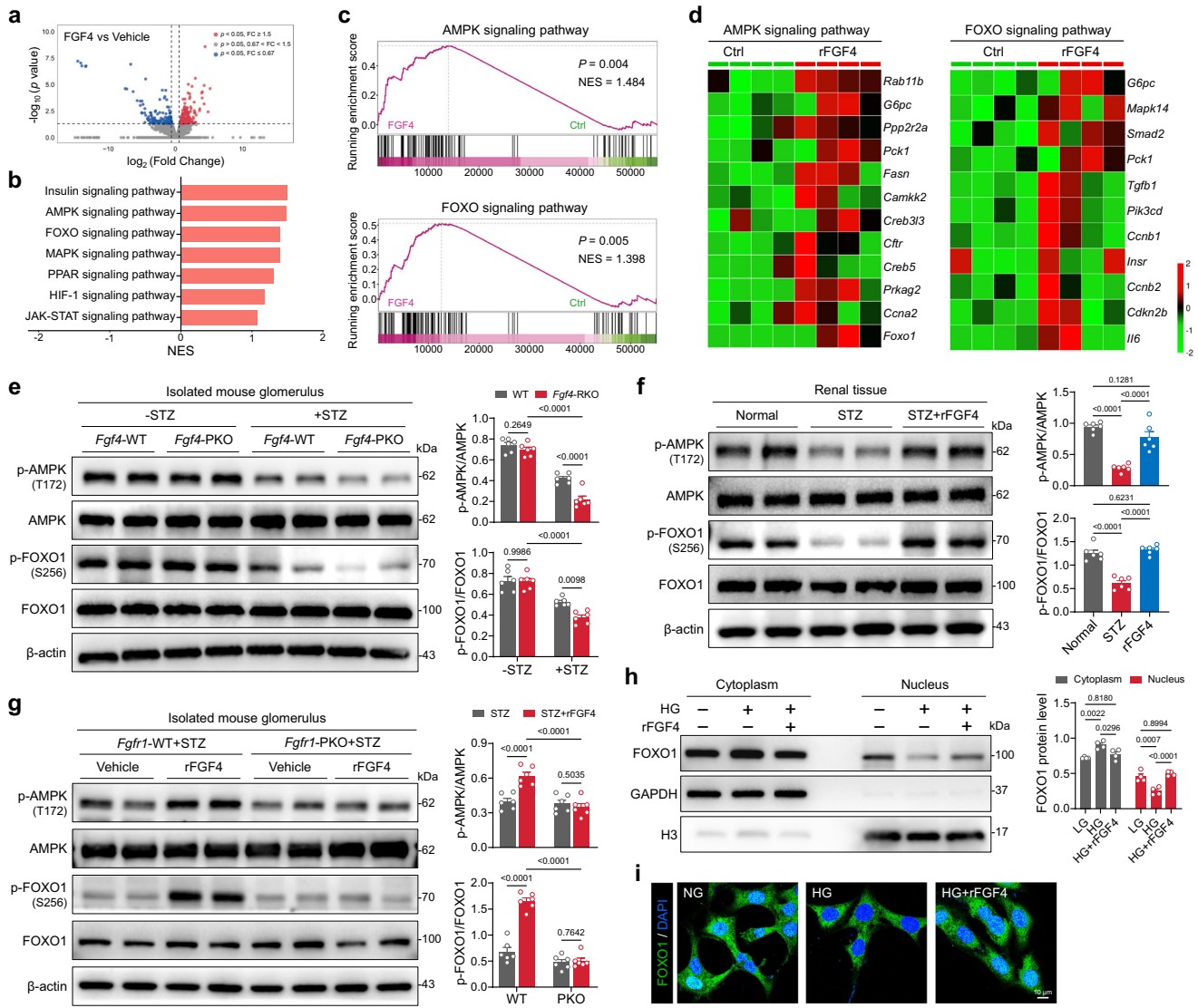

**Fig. 5 | Activation of the AMPK-FOXO1 pathway mediates the protective effects of FGF4 in DKD. a** Volcano plots depicting differentially expressed genes in the kidney of rFGF4- and vehicle-treated mice. Red dots represent upregulated genes ($p < 0.05$, FC $\geq 1.5$); blue dots represent downregulated genes ($p < 0.05$, FC $\leq 0.67$); grey dots represent genes with no significant change ($p > 0.05$, $0.67 <$ FC $< 1.5$). $n = 4$ per group. **b, c** KEGG pathway enrichment analysis showing the involvement of apoptosis and oxidative stress pathways in the kidneys of the indicated groups based on RNA-seq data. **d** Heat map of differentially expressed genes involved in the AMPK (left) and FOXO1 (right) pathways. **e** Western blot analysis of proteins involved in the AMPK–FOXO1 signaling cascade in total renal lysates from WT and *Fgf4*-PKO diabetic or normal mice ($n = 6$). **f** Immunoblotting analysis of AMPK and FOXO1 pathway activation in total renal lysates from STZ-induced diabetic mice treated with rFGF4 or vehicle control ($n = 6$). Phosphorylated protein levels are expressed relative to total protein levels and presented as scatter plots. **g** Western

blot analysis and quantification of phosphorylated (p)-AMPK and p-Foxo1 levels in isolated glomeruli from each treatment group ($n = 6$). **h** Effect of rFGF4 on Foxo1 localization in the cytoplasm or nucleus of MPC-5 cells as determined by subcellular fractionation analysis. H3 and GAPDH were used as controls for nuclear and cytosolic fractions, respectively ($n = 4$). **i** Immunofluorescence analysis of the effect of AMPK knockdown on Foxo1 (green) subcellular localization in MPC-5 cells. Nuclei were counterstained with DAPI (blue). Representative images from three independent experiments with similar results. β-actin was used as a loading control. Data are presented as mean ± s.e.m. ***$p < 0.001$ as determined by ordinary two-way ANOVA followed by Sidak's multiple comparisons tests (**e**–**h**) or ordinary one-way ANOVA followed by Tukey's multiple comparisons test (**f**); ns, not significant. KEGG, Kyoto Encyclopedia of Genes and Genomes; NES, Normalized Enrichment Score; FC, Fold change; H3, Histone 3.

To further elucidate the role played by AMPK-FOXO1 in mediating the therapeutic effects of FGF4 on DKD, we established a mouse model with podocyte-specific knockout of *Ampka1/a2* (*Ampk*-PKO) and subjected these mice to STZ-induced diabetes, followed by treatment with rFGF4 or vehicle control (Fig. 6a, b and Supplementary Fig. 11a). Reduction in the renal injury marker UACR induced by rFGF4 in diabetic mice was markedly abolished in *Ampk*-PKO diabetic mice (Fig. 6c). Histological analysis and Nephrin immunohistochemistry revealed substantial improvement in renal pathology and podocyte loss in diabetic *Ampk*ᶠˡ/ᶠˡ mice treated with rFGF4

compared to vehicle-treated controls. However, these therapeutic effects were absent in diabetic *Ampk*-PKO mice (Fig. 6d, e). Similarly, the ability of rFGF4 to preserve podocyte structure in diabetic mice was blunted by the loss of *Ampka1/a2* (Fig. 6f, g). Finally, *Ampk* deficiency prevented rFGF4 from reducing glomerular oxidative stress and apoptosis, indicating the essential role of AMPK in podocytes for rFGF4 to protect against diabetic renal injury (Fig. 6h, i). Consistent with these in vivo analyses, *Ampk* knockdown in MPC-5 cells negated the ability of rFGF4 to reduce apoptosis and ROS accumulation, as well as upregulate anti-oxidative stress and anti-

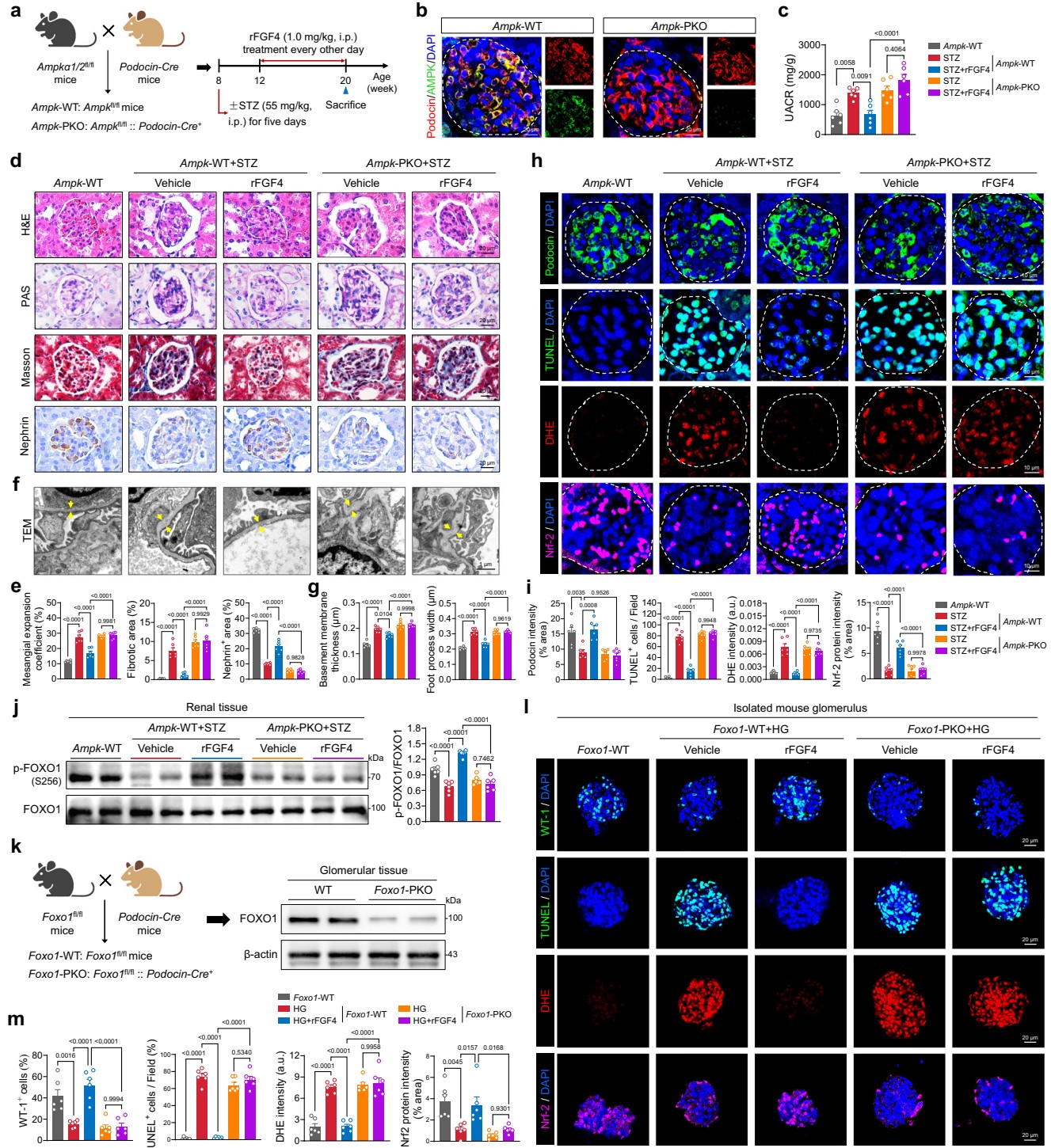

**Fig. 6 | AMPK-FOXO1 dependency in the therapeutic effects of rFGF4 against renal injury. a** Schematic of the rFGF4 treatment regimen for WT and podocyte-specific *Ampk* knockout (*Ampk*-PKO) mice with STZ-induced diabetes. Male mice (6 weeks old) received i.p. injections of STZ (55 mg/kg body weight) for five consecutive days. At 12 weeks of age, these mice received i.p. injections of rFGF4 (1.0 mg/kg body weight) or PBS vehicle every other day for 8 weeks. **b** Immunofluorescence staining showing co-localization of AMPK (green) with podocin (red) in glomeruli. **c** Quantification of UACR in the indicated groups ($n = 6$). **d, e** Representative images and quantification of renal sections stained with H&E, PAS, Masson's trichrome, and Nephrin ($n = 6$). **f, g** TEM images and quantification of renal tissues from the indicated groups, with yellow arrows indicating GBM width ($n = 6$). **h, i** Representative images and quantification of podocin (green), TUNEL

(green), DHE (red), and Nrf-2 (pink) staining in renal sections of WT and *Ampk*-PKO diabetic mice after rFGF4 treatment ($n = 6$). **j** Western blot analysis of phosphorylated (p)-FOXO1 and total FOXO1 in renal lysates from WT and *Ampk*-PKO mice under diabetic or normal conditions ($n = 6$). **k** Schematic of podocyte-specific *Foxo1* knockout (*Foxo1*-PKO) and Western blot analysis of FOXO1 expression in glomeruli isolated from *Foxo1*-PKO and WT control mice. **l, m** Representative images and quantification of WT-1 (green), TUNEL (green), DHE (red), and Nrf-2 (pink) staining in renal sections of WT and *Foxo1*-PKO diabetic mice ($n = 6$). Nuclei were counterstained with DAPI (blue). Data are presented as mean ± s.e.m. $^*p < 0.01$, $^{**}p < 0.01$, $^{***}p < 0.001$, $^{****}p < 0.0001$ as determined by ordinary one-way ANOVA followed by Tukey's multiple comparisons test (**c**–**m**); ns, not significant.

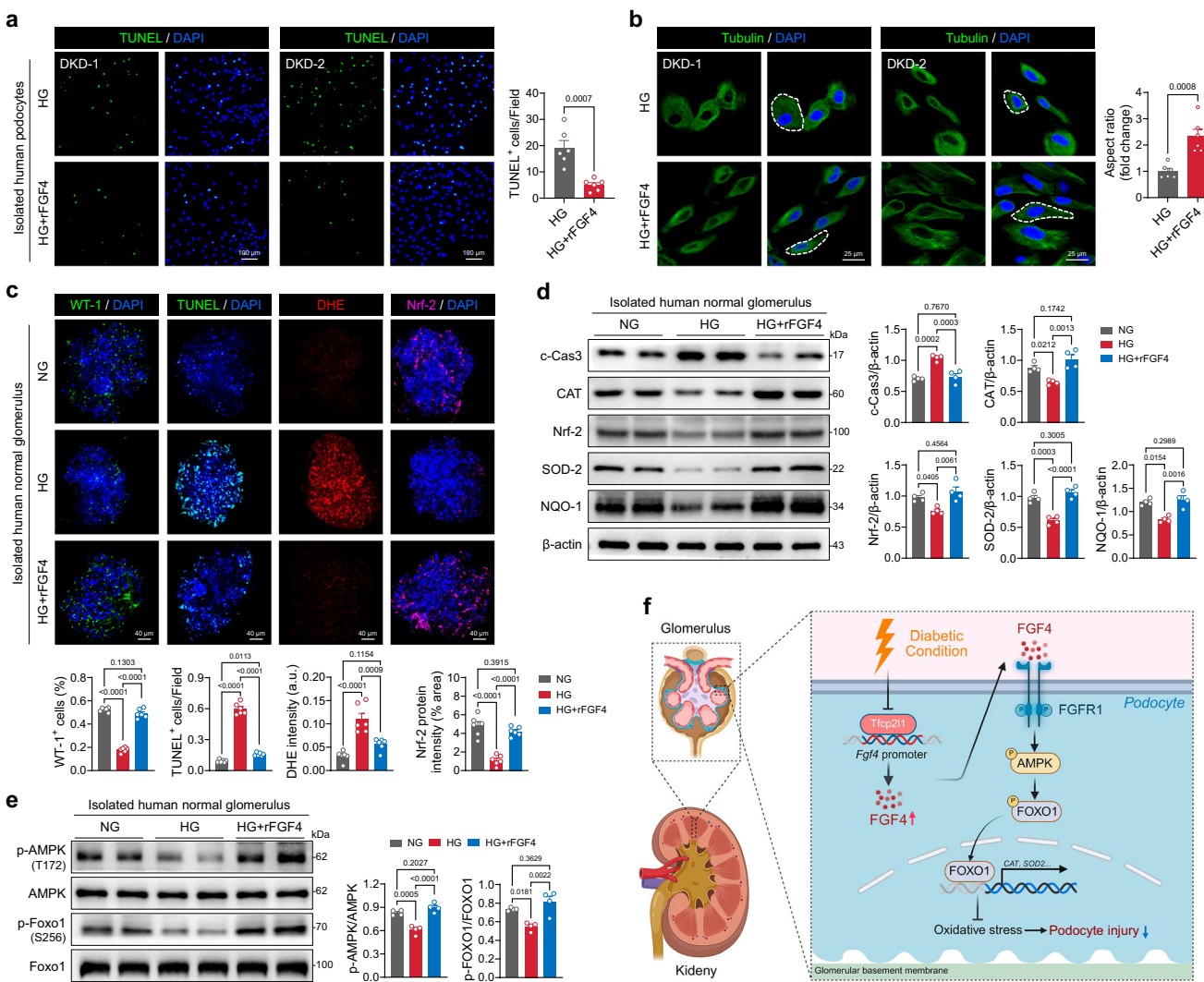

**Fig. 7 | Clinical relevance and therapeutic potential of rFGF4 in DKD. a, b** Human podocytes were isolated from the urine of DKD patients and exposed to high glucose conditions with or without rFGF4 treatment. **a** Representative images of TUNEL staining (green) and quantification of TUNEL-positive cells from the indicated groups ($n = 6$). **b** Immunofluorescence analysis of tubulin (green) and quantification of the aspect ratio (width to height ratio) of human podocytes ($n = 6$). White dashed lines mark the edge of the cells. **c** Representative images and quantification of isolated human glomeruli stained for WT-1 (green), TUNEL (green), DHE (red), and Nrf-2 (pink). Nuclei were counterstained with DAPI (blue). **d** Western blot and quantitative analysis of c-Cas3, CAT, SOD-2, and Nrf-2 protein levels in human glomeruli from the indicated groups ($n = 4$). β-actin served as a loading control. **e** Western blot analysis of the activation of AMPK-FOXO1 pathway components in total glomerular tissue lysate. The ratios of phosphorylated proteins to total proteins are shown as scatter plots ($n = 4$). **f** Schematic showing that FGF4 protects against DKD by activating the FGFR1-AMPK-FOXO1 signaling axis. Data are presented as mean ± s.e.m. $^{*}p < 0.05$, $^{**}p < 0.01$, $^{***}p < 0.001$, $^{****}p < 0.0001$ as determined by unpaired two-tailed Student's $t$-test (**a**, **b**) or ordinary one-way ANOVA followed by Tukey's multiple comparisons test (**c**–**e**); ns, not significant. c-Cas3, cleaved-Caspase3; CAT, Catalase.

apoptotic proteins (Supplementary Fig. 12). Furthermore, *Ampk*-PKO abolished the rFGF4-induced upregulation of FOXO1 phosphorylation (Fig. 6j).

Given that FOXO1 is a key downstream regulatory factor of AMPK, we next generated podocyte-specific *Foxo1* knockout mice (*Foxo1*-PKO) and isolated glomeruli for high-glucose modeling and rFGF4 treatment (Fig. 6k and Supplementary Fig. 11b). The loss of FOXO1 blunted the abilities of rFGF4 to increase podocyte numbers, reduce apoptosis, and decrease ROS accumulation (Fig. 6l, m). Collectively, these data highlighted the essential role of the AMPK-FOXO1 pathway in podocytes in mediating the therapeutic benefits of rFGF4 in DKD.

### Clinical translational potential of rFGF4 in DKD

To assess the clinical potential of rFGF4 in treating DKD, shed podocytes in the urine of patients diagnosed with DKD (Supplementary Table 1) were collected and exposed to rFGF4. In alignment with our mouse and cell culture experiments, rFGF4 reduced high glucose-induced apoptosis in human podocytes (Fig. 7a). Under high-glucose conditions, the morphological transition of podocytes from a spindle shape to an elliptical shape, a characteristic feature of DKD, was reversed by rFGF4 treatment (Fig. 7b). Finally, we obtained human glomeruli from kidneys subjected to surgical resection and exposed them to high glucose conditions. rFGF4 treatment effectively alleviated high glucose-induced podocyte loss, apoptosis, and ROS accumulation, supporting our data from renal glomeruli of diabetic mice (Fig. 7c). Consistent with these morphological results, rFGF4 treatment induced the downregulation of c-Cas3 and the upregulation of Nrf-2 and SOD2 in human renal glomeruli (Fig. 7d). Significantly, rFGF4 treatment increased the levels of p-AMPK and p-FOXO1 in isolated human renal glomeruli (Fig. 7e). Collectively, these data demonstrate that FGF4 is a promising therapeutic candidate for treating DKD, and

that these beneficial effects are mediated by activation of the FGFR1-AMPK pathway.

We next examined whether current first-line therapies for DKD—renin-angiotensin system (RAS) blockade and sodium-dependent glucose transporter 2 (SGLT2) inhibition—modulate podocyte expression of FGF4 and FGFR1. In the STZ-induced diabetic mouse model, we administered either the RAS blocker losartan potassium (LOS) or the SGLT2 inhibitor empagliflozin (EMP). Blood glucose levels were significantly reduced in EMP-treated mice but remained unchanged with LOS treatment (Supplementary Fig. 13a). Compared to the STZ control group, LOS treatment was associated with increased glomerular expression of FGF4 at both the protein and mRNA levels (Supplementary Fig. 13b, d), suggesting that its podocyte-protective effects may be partially mediated via upregulation of FGF4. In contrast, EMP treatment did not significantly alter renal FGF4 expression relative to the STZ controls (Supplementary Fig. 13b, e). Neither treatment significantly affected FGFR1 expression levels (Supplementary Fig. 13c, e).

## Discussion

DKD remains one of the leading causes of end-stage renal disease worldwide, with podocyte loss playing a pivotal role in its pathogenesis[33,37]. Despite advances in the management of diabetes, current DKD therapies primarily slow disease progression rather than offering curative solutions, underscoring the urgent need for more effective treatment strategies that target underlying mechanisms[2,4,34]. In this study, we present evidence supporting the therapeutic potential of rFGF4 in halting DKD progression. We demonstrate that endogenous FGF4 expression is significantly reduced in both murine and human renal tissues during DKD progression, with podocyte-specific knockout of *Fgf4* exacerbating kidney damage. In contrast, supplying exogenous rFGF4 mitigated DKD progression primarily by inhibiting oxidative stress to promote podocyte survival and glomerular function via the FGFR1-AMPK-FOXO1 signaling axis (Fig. 7f). Importantly, rFGF4 restored the cytoskeletal integrity of primary human podocytes derived from healthy donors and mitigated injury to human glomerular podocytes under high glucose conditions. These data highlight the promising potential for translating rFGF4 to the clinic.

Recent studies have characterized the critical roles played by FGFs in diabetic complications, including DKD. Early research centered on endocrine FGF21, which protects against diabetic nephritis, oxidative stress, fibrosis, and lipid accumulation[38,39]. Our previous research has demonstrated that paracrine FGF1 is expressed in a diverse array of kidney cells, including mesangial cells and podocytes, with its expression markedly downregulated in DKD. Additionally, exogenous administration of FGF1 primarily mitigates inflammatory responses, consequently ameliorating the condition of DKD[40]. Another study implicated FGF13 and FGF11 in DKD exacerbation, with FGF13 promoting endothelial cell dysfunction and FGF11 contributing to mesangial cell proliferation and fibrosis[41,42]. However, the roles of FGF family members in podocytes and the associated molecular mechanisms remain poorly understood. In this study, we identified that FGF4 is predominantly expressed by podocytes of the glomerulus, and that deletion of *Fgf4* specifically in podocytes exacerbated DKD. Although single-cell sequencing revealed relatively low levels of *Fgf4* expression in these cells[43–45], our findings illustrate that FGF4 serves as a high-potency, low-abundance molecular guardian in podocyte homeostasis. In addition, FGFR1, the receptor through which FGF4 regulates metabolic homeostasis[28], also localized to glomerular podocytes. This observation is in agreement with previous single-cell sequencing analysis of mouse podocytes[46]. Upon receiving the FGF4 signal, FGFR1 activates the AMPK-FOXO1 signaling pathway to reduce podocyte apoptosis and oxidative stress, thereby alleviating DKD.

Oxidative stress, characterized by excessive ROS accumulation, plays a pivotal role in the progression of DKD[47,48]. Podocytes, essential for maintaining the glomerular filtration barrier and renal integrity, are particularly susceptible to long-term hyperglycemia, which leads to ROS accumulation and apoptosis[49,50]. Our study highlights the protective role of rFGF4 in reducing ROS accumulation, preserving podocyte integrity, and mitigating kidney damage. Notably, while rFGF4 reduced blood glucose levels in *db/db* diabetic mice, it did not affect blood glucose levels in the STZ-induced model. The lack of a hypoglycemic effect of FGF4 in type 1 diabetic mice may be due to its inability to substitute for insulin across glucose-regulating tissues (e.g., muscle, liver) and the potential suppression of signaling pathways under an insulin-deficient metabolic state[28]. In both DKD mouse models, rFGF4 alleviated renal dysfunction, pathological changes, fibrosis, and podocyte injury, confirming its role in mitigating DKD via non-glycemic-dependent mechanisms. Structural changes observed in the kidneys, particularly in glomerular architecture, further emphasize the role of FGF4 in maintaining renal structural integrity.

AMPK functions as an energy sensor that is activated under conditions of low energy. Activated AMPK promotes ATP production and inhibits energy-consuming processes[51,52]. During oxidative stress, AMPK activation plays a pivotal role in enhancing cellular antioxidant defense mechanisms, thereby improving cell survival[16]. In the kidney, AMPK activation reduces podocyte permeability to albumin and alleviates podocyte dysfunction[53,54]. These findings highlight the therapeutic potential of AMPK activators in managing renal diseases, suggesting that targeting AMPK may be a promising strategy for improving renal health. FOXO1, on the other hand, is a transcription factor that regulates a wide variety of cellular processes, including the cell cycle, apoptosis, and responses to oxidative stress[55,56]. AMPK can activate FOXO1, leading to the transcriptional upregulation of genes that enhance the antioxidant response[36]. This interplay between AMPK and FOXO1 helps maintain cellular integrity under stress. Our research demonstrates that upon receiving the FGF4 signal, FGFR1 on podocytes activated the AMPK-FOXO1 signaling cascade, which subsequently mitigated oxidative stress and reduced podocyte apoptosis. These findings align with our previous studies demonstrating that the effects of FGF4 on both central glucose regulation and non-alcoholic fatty liver disease amelioration are mediated through FGFR-dependent AMPK activation in target cells[26,27], thereby establishing AMPK as a center molecular node within FGF4's biological network. Genetic knockout of AMPK or FOXO1 abrogated the protective effects of rFGF4 against diabetes-induced renal injury, underscoring the critical role of the AMPK-FOXO1 axis in mediating the therapeutic effects of FGF4 in DKD.

In summary, our study demonstrates that podocyte-derived FGF4 activates the FGFR1-AMPK-FOXO1 signaling pathway to mitigate oxidative stress and promote podocyte survival, thereby preserving glomerular integrity and protecting the kidney from diabetes-induced injury. These findings expand the horizon of the physiopathological roles of FGF4 in renal homeostasis and shed new light on potential strategies for pharmacotherapeutic intervention in DKD.

## Methods
### Human renal samples
Renal biopsies from patients with chronic kidney disease had been performed as part of routine clinical diagnostic investigation. The samples of renal biopsies were obtained from the Department of Nephrology, The Second Affiliated Hospital and Yuying Children's Hospital of Wenzhou Medical University. Control samples were obtained from the healthy kidney poles of individuals who underwent tumor nephrectomies without diabetes or chronic renal disease. Written informed consent to participate was obtained from all human participants. None of the human study participants received compensation. The detailed information of the human renal biopsy samples is provided in Supplementary Table 1. The entire human study, encompassing the experimental design, methodologies, and data analysis procedures, were approved by the Ethics Committee of The Second Affiliated Hospital and Yuying Children's Hospital of Wenzhou

Medical University, as well as The Second Affiliated Hospital of Zhejiang University. All experimental procedures were conducted in accordance with the Declaration of Helsinki on ethical principles for medical research involving human participants.

## Expression and purification of recombinant FGF4

N-terminally truncated human FGF4 (rFGF4) (Ala67-Leu206) were expressed in *E. coli* (BL21) and purified by using a heparin column followed with a size-exclusion chromatography column according to published protocols[25,26].

## Extraction and cultivation of primary glomeruli

Normal human renal tissues used in this study were obtained from regions distant from the diseased site following surgical resection of renal tumors. Glomeruli were isolated through a differential screening method with 250 μm, 180 μm, and 100 μm cell sieves. For mouse primary glomeruli, kidney samples were isolated using differential sieving with 100 μm, 70 μm, and 40 μm cell sieves. Glomeruli that sieved through the 40 μm filter were collected, appropriately suspended, and plated onto polylysine-coated slides to create a monolayer structure. Cultures were maintained in RPMI 1640 medium supplemented with glutamine (5 mmol/L), n-2-hydroxyethylpiperazine-n′-2-ethanesulfonic acid (HEPES) (15 mmol/L), penicillin (100 U/ml), and streptomycin (100 μg/ml), under conditions of 5% $CO_2$ and 95% air. *Fgf4*-PKO and *Fgfr1*-PKO mice received chronic rFGF4 treatment, after which glomeruli were isolated and fixed in 4% paraformaldehyde (PFA) for 30 min for subsequent immunofluorescence analyses.

## Establishment of DKD pathological model using primary glomeruli under high-glucose conditions

Primary glomeruli isolated from *Foxo1*-PKO mice or human specimens were suspended in RPMI 1640 medium supplemented with glutamine (5 mmol/L), HEPES (15 mmol/L), penicillin (100 U/mL), and streptomycin (100 μg/mL) to support cell growth. Based on the experimental design, the glomeruli were divided into three groups: control group, high-glucose group, and treatment group.

For the control group, the medium was supplemented with 5 mM glucose to maintain normal glucose concentrations. For the high-glucose group, the medium was supplemented with 33 mM glucose to simulate a hyperglycemic environment. For the treatment group, the high-glucose medium was supplemented with rFGF4 (200 ng/mL). All groups were cultured in a 37 °C, 5% $CO_2$ incubator for 36 h to mimic the pathological conditions of diabetic kidney disease.

## Extraction and cultivation of human primary podocytes

Human urine samples were obtained from two groups of diabetic patients: non-DKD controls (defined as clinically diagnosed diabetic patients with persistently normal renal functional indicators) and diagnosed DKD patients. These samples were centrifuged at $400 \times g$ for 5 min, and the resulting sediment was washed two times with phosphate-buffered saline (PBS). For cell culture, 12-well plates were precoated with type I collagen at a concentration of 50 μg/ml. The pelleted cells were suspended in DMEM/F12 medium supplemented with 10% fetal bovine serum (FBS) and 1% insulin-transferrin-selenium-ethanolamine (ITS-X). Cultivation was maintained at a constant temperature of 37 °C, with medium replenishment every 2 days based on cell status. Cell collection occurred approximately on day 10.

## Animal study experimental protocol

Male C57BL/6 J, *db/db* (C57BLKS/J-lepr$^{db}$/lepr$^{db}$), and their nondiabetic *db/dm* littermates (8–10 weeks old) were purchased from the Model Animal Research Center of Nanjing University (Nanjing, China). *Fgf4*-floxed mice (*Fgf4*$^{flox/flox}$, Cat. No. T009264) were obtained from Gem-Pharmatech Co., Ltd. (Jiangsu, China), while *Fgfr1*-flox mice (*Fgfr1*$^{flox/flox}$) and *podocin (Pod)*-Cre mice were acquired from Cyagen Biosciences

Inc. (Suzhou, China). *Foxo1*-Flox mice (*Foxo1*$^{flox/flox}$, Cat. No. NM-CKO-200177) were purchased from Shanghai Model Organisms Center (Shanghai, China). *Ampkα1/α2*-floxed (*Ampkα1/α2*$^{flox/flox}$) mice were generously provided by Dr. Sheng-Cai Lin (Xiamen University, Fujian, China). *Fgfr1*$^{flox/flox}$, *Ampkα1/α2*$^{flox/flox}$, and *Foxo1*$^{flox/flox}$ mice were separately bred with *Pod*-Cre transgenic mice, resulting in the development of podocin-specific *Fgfr1* knockout (*Fgfr1*$^{flox/flox}$; *Pod*-Cre$^+$, *Fgfr1*-PKO), *Ampkα1/α2* double knockout (*Ampk*$^{flox/flox}$; *Pod*-Cre$^+$, *Ampk*-PKO), and *Foxo1* knockout (*Foxo1*$^{flox/flox}$; *Pod*-Cre$^+$, *Foxo1*-PKO) mice. All animal protocols were approved by the Institutional Animal Care and Use Committee at Wenzhou Medical University. Prior to experimentation, animals were acclimated to laboratory conditions and were housed in a controlled environment ($22 \pm 2$ °C, 50–60% humidity, and 12 h light/dark cycle with lights on at 7:00 AM). Throughout the study, animals were provided with standardized commercial maintenance diet (Sibefu; Product No. mk032) and purified water ad libitum.

## Generation of podocyte-specific *Fgfr1*, *Ampkα1/α2*, and *Foxo1* knockout mice

Mice with specific knockout of the *Fgfr1*, *Ampkα1/α2*, and *Foxo1* genes in glomerular podocytes were generated employing the Cre-loxP system. Age-matched control mice lacking Cre (*Pod*-Cre-/*Fgfr1*$^{flox/flox}$, *Pod*-Cre-/*Ampkα1/α2*$^{flox/flox}$, and *Pod*-Cre-/*Foxo1*$^{flox/flox}$) were utilized for comparison. Genomic DNA extracted from mouse tails served as the substrate for mouse genotyping conducted through PCR. The specific primers employed for genotyping are as follows: *Ampkα1*-flox-F, 5′-CCCACCATCACTCCATCTCT-3′; *Ampkα1*-flox-R, 5′-AGCCTGCTTGG CACACTTAT-3′; *Ampkα2*-flox-F, 5′-GCAGGCGAATTTCTGAGTTC-3′; *Ampkα2*-flox-R, 5′-TCCCCTTGAACAAGCATACC-3′; *Fgfr1*$^{flox}$-F, 5′-GGA CTGGGATAGCAAGTCTCTA-3′; *Fgfr1*$^{flox}$-R, 5′-GTGGATCTCTGTGAGCC TGAG-3′; *Foxo1*$^{flox}$-F, 5′-GGACTGGGATAGCAAGTCTCTA-3′; and *Foxo1*$^{flox}$-R, 5′-GGACTGGGATAGCAAGTCTCTA-3′. Genotyping of mice expressing *Pod*-Cre was performed with the following primers: *Pod*-Cre-F, 5′-CCTGGAAAATGCTTCTGTCCG-3′; and *Pod*-Cre-R, 5′-CAGGG TGTTATAAGCAATCCC-3′.

In addition to spontaneously induced DKD in *db/db* mice via prolonged exposure to high glucose, DKD was induced in other mouse types (C57BL/6, *Fgf4*-floxed, *Fgf4*-RKO, *Fgfr1*-floxed, *Fgfr1*-PKO, *Ampkα1/α2*-floxed, and *Ampkα1/α2*-PKO) using streptozotocin (STZ, S0130, Sigma-Aldrich, St. Louis, MO, USA). Briefly, 8 week-old male mice were intraperitoneally injected with STZ (50 mg/kg; dissolved in 100 mmol/L citrate buffer, pH 4.5) or citrate buffer alone as a control for 5 consecutive days. Subsequently, these mice were allowed free access to food and water for 8 weeks to induce the DKD model. For drug treatment, human recombinant N-truncated rFGF4 (1.0 mg/kg body weight) or PBS vehicle was intraperitoneally injected into mice every other day for an additional 8 weeks. Body weight and blood glucose levels were monitored every two days. Following the final injection of rFGF4 or PBS, urine samples was collected for 24 h using metabolic cages (TSE Systems, MO, USA). The mice were then humanely euthanized, followed by the collection of blood samples and kidney tissues for comprehensive analyses. Plasma glucose levels were measured using a FreeStyle complete blood glucose monitor (Abbott Diabetes Care Inc., Alameda, CA, USA). Serum levels of BUN (C013−2, Jiancheng, Nanjing, China), albuminuria (C035−2, Jiancheng, Nanjing, China), and creatinine (C011−2, Jiancheng, Nanjing, China) were quantified using assay kits according to the manufacturer's instructions.

## RNA-Scope in situ hybridization

*Fgf4* mRNA expression levels were assessed using the RNA-Scope assay (Bio-Techne Corporation, Cat. No. 322350) following the manufacturer's instructions. Briefly, kidney tissues were harvested following PBS cardiac perfusion, after which renal sections were deparaffinized and treated with $H_2O_2$ for 10 min at room temperature to expose RNA. Following PBS washes, sections were immersed in Target Reagent

solution for 30 min and subsequently incubated with Protease Plus at 40 °C for 30 min. Following another wash, renal sections were incubated with either a human FGF4 probe (Bio-Techne Corporation, Cat. No. 512291) or a mouse *Fgf4* probe (Bio-Techne Corporation, Cat. No. 514311) for 2 h at 40 °C, and they were subsequently reacted with Amp1 - 6 reagent and RED working solution. Fluorescence images were visualized and captured using a Nikon C2si Confocal Microscope.

## Histological analysis of renal tissues

Tissues were fixed by incubation with 4% PFA at 4 °C overnight. The fixed tissues were then embedded in paraffin and cross-sectioned (4 μm) for histological examination. H&E, PAS, and Masson's trichrome staining were conducted following the manufacturers' instructions (Solarbio, Beijing, China). For each human subject or mouse, at least six randomly chosen fields within each section were captured using a Leica DM750 P microscope (Leica, Germany) at 20× or 40× magnification. The quantification of the mesangial amplification coefficient after H&E staining was performed using ImageJ software (National Institutes of Health, Bethesda, USA) on randomly selected regions (40×). The data are expressed as the ratio of the area of the renal capsule to the glomerular cluster. Quantification of collagen content after Masson's trichrome staining was performed by analyzing the percentage of staining area in randomly selected fields (40×) using ImageJ software (National Institutes of Health, Bethesda, USA). The data are expressed as the positive stained area versus the total analyzed area.

## Cell culture and siRNA transfection

The conditionally immortalized mouse podocyte cell line (MPC-5), obtained from Procell (Wuhan, China; Cat. No. CL-0855), was cultured in RPMI 1640 medium containing 10% FBS, 100 U/ml penicillin, and 100 μg/ml streptomycin. All cells were cultured at 37 °C and 5% $CO_2$. To silence the *Fgf4*, *Ampk*, *Fgfr1*, and *Foxo1* genes, MPC-5 cells were transfected with mouse *Fgf4* siRNA (si*Fgf4*; Cat. No. sc-39451; Santa Cruz), mouse *Ampkα1/2* siRNA (si*Ampk*; Cat. No. sc-45313; Santa Cruz) and mouse *Fgfr1* siRNA (si*Fgfr1*; Cat. No. sc-29317; Santa Cruz) respectively. For control cells, MPCs were transfected with control siRNA (control siRNA; Cat. No. sc-37007; Santa Cruz). Cells were plated in 6-well plates at 60−70% confluency and transfected with 5 μL of Lipofectamine 3000 (Thermo Fisher, Invitrogen) transfection reagent and 50 nM siRNA. For some experiments, at 24 hours post-transfection, cells were incubated with 33.3 mM high glucose or 100 ng/ml rFGF4 for 24 hours. Transfection efficiency was evaluated via Western blot analysis, and cells were assessed for apoptosis- and oxidative stress-related factors following transfection.

## Dual-luciferase reporter gene

According to the instructions provided by Hanbio (Hangzhou, China), the FGF4 promoter-driven luciferase reporter plasmid, pGL3-basic-*FGF4*-WT, was constructed utilizing the pGL3 luciferase vector. HEK-293T cells were plated in 24-well plates, and co-transfected with pGL3-basic-*FGF4*-WT, pcDNA3.1-*Tfcp2l1*, and the internal control pRL-TK plasmid. Following a 48 hour transfection period, luciferase activity was measured using the Dual-Luciferase Reporter Assay Kit (Product No. HB-DLR-100) from Hanbio, following the manufacturer's protocol. The Firefly luciferase activity was normalized to the Renilla luciferase activity (Firefly-to-Renilla ratio) to evaluate the transcriptional activity of the FGF4 promote. All experiments were conducted in triplicate to ensure data reliability and reproducibility.

## Western blotting

Renal tissues (30-50 mg) or cells were homogenized in lysis buffer (25 mM Tris, pH 7.6; 150 mM NaCl; 1% NP-40, 1% sodium deoxycholate; and 0.1% SDS) containing protease and phosphatase inhibitors (Applygen, P1260-1). The cell or tissue lysates were mixed with SDS sample buffer, boiled, and separated on 8-15% SDS-PAGE, followed by electrophoretic transfer to a nitrocellulose membrane. The membranes were blocked with 10% nonfat milk in TBST for 2 h, followed by incubation with primary antibodies at 4 °C overnight. The following primary antibodies were used: FGF4 (1:1000, Cat. No. ab106355, Abcam), β-actin (1:1000, Cat. No. HC201-02, TransGen Biotech), Nephrin (1:1000, Cat. No. AF7951, Affinity), Podocin (1:1000, Cat. No. sc518088, Santa Cruz), TGF-β1 (1:1000, Cat. No. ab215715, Abcam), cleaved caspase3 (1:1000, Cat. No. TA7022S, Abmart), Bax (1:1000, Cat. No. sc493, Santa Cruz), Nrf-2 (1:5000, Cat. No. 16396, Proteintech), HO-1 (1:2000, Cat. No. ab13243, Abcam), FGFR1 (1:1000, Cat. No. 9740S, Cell Signaling), phospho-AMPKα (1:1000, Cat. No. 2535, Cell Signaling), AMPKα (1:1000, Cat. No. ab32047, Abcam), FOXO1 (pS256) (1:1000, Cat. No. PA5293, Abmart), FoxO1a (1:1000, Cat. No. T55376, Abmart), catalase (1:1000, Cat. No. A11777, ABclonal), SOD2 (1:1000, Cat. No. PK08370, Abmart), NQO-1 (1:1000, Cat. No. 67240, Proteintech), Bcl-2 (1:1000, Cat. No. T40056, Abmart), phospho-ACC (1:1000, Cat. No. 11818, Cell Signaling), and Histone H3 (1B1B2) (1:1000, Cat. No. 14269, Cell Signaling). Proteins bands were visualized using a horseradish peroxidase-conjugated secondary antibody and enhanced chemiluminescence reagent (TransGen Biotech, DW101-01). Densitometric analysis was performed using ImageJ software (version 1.44p, NIH).

## TUNEL assay

Apoptotic cells were detected via the TUNEL assay (Elabscience; One-step TUNEL Assay Kit; E-CK-A320) following the manufacturer's instructions. The number of TUNEL-positive cells was evaluated in six fields per section by using a fluorescence inverted microscope (Nikon, Japan).

## Dihydroethidium (DHE) assay

Cellular ROS levels were detected using the DHE assay (Beyotime; Cat. No. S0063) according to the manufacturer's protocol. In brief, the cells were incubated with PBS containing 5 μM DHE at 37 °C for 30 min. ROS-positive cells were counted using a fluorescence inverted microscope at a wavelength of 535 nm (Nikon, Japan).

## Immunohistochemistry

Mouse kidneys were fixed in 4% PFA, embedded in paraffin, and sectioned at a thickness of 5 μm. The renal sections were heated at 65 °C for 5 h, followed by deparaffinization with xylene and rehydration through a series of ethanol solutions of decreasing concentrations. After washing the renal sections with PBS, they were boiled in 10 mM sodium citrate buffer (pH 6.0) at 100 °C for 2 min. After two washes in PBS, the renal sections were incubated with 3% $H_2O_2$ for 30 min and then blocked with 5% BSA for 1 h. The sections were then incubated overnight at 4 °C with primary antibodies against Wilms Tumor (1:50, Cat. No. ab89901, Abcam), Nephrin (1:1000, Cat. No. ab216341, Abcam), NPHS2 (1:300, Cat. No. ab181143, Abcam), collagen IV (1:400, Cat. No. ab6586, Abcam), and Ki67 (1:2000, Cat. No. 28074-1-AP, Proteintech). Following a PBS wash, the sections were incubated with HRP-conjugated goat anti-rabbit secondary antibody (Jackson ImmunoResearch Labs, Cat. No. 11-035-003) or goat anti-mouse IgG (H + L) HRP (Bioworld, Cat. No. BS12478) for 1 h at room temperature. The sections were then developed with a DAB kit (ZSGB-BIO, Cat. No. ZLI-9018) and counterstained with hematoxylin according to the manufacturer's protocols. The images were captured using a microscope (Nikon, Tokyo, Japan).

## Immunofluorescence staining

After antigen retrieval, paraffin-embedded renal sections were washed twice with PBS, blocked with 5% BSA for 1 h, and then incubated overnight at 4 °C with primary antibodies against FGF4 (1:400, Cat. No. 106355, Abcam), Podocin (1:100, Cat. No. sc518088, Santa Cruz), FoxO1a (1:200, Cat. No. T55376, Abmart), CD31 (1:200, Cat. No. 66065, Proteintech), β-Tubulin (C66) (1:1000, Cat. No. M20005, Abmart), FGFR1 (1:300, Cat. No. ab824, Abcam), Alpha-smooth muscle actin

(1:300, Cat. No. 124964, Abcam), Wilms Tumor (1:50, Cat. No. ab89901, Abcam), AQP-1 (1:250, Cat. No. AF5231, affinity) and Nrf-2 (1:200, Cat. No. ab62352, Abcam). The sections were then washed three times with PBS and incubated with Alexa Fluor 488- or 647-conjugated secondary antibodies (1:300, Cat. No. ab150073 or ab150115, Abcam) for 1 h. The sections were then incubated with DAPI (Southern Biotech, Birmingham) for 10 min. Co-staining for FGFR1 (1:1000, Cat. No. 60325, Proteintech) and NPHS2 (1:1000, Cat. No. 20384, Proteintech) was performed using a multiplex staining kit (Cat. No. AFIHC034, AiFang Biological) based on Tyramide Signal Amplification (TSA) technology. Immunofluorescence was analyzed using a confocal microscope (Leica, Mannheim, Germany).

### RNA isolation and real-time PCR
Total RNA was extracted from the renal cortex using TRIzol reagent (TransGen Biotech, ET111-01), and the RNA was reverse transcribed to synthesize cDNA using a SuperScript™ II Reverse Transcriptase kit (TransGen Biotech, AT341) following the manufacturer's instructions. Quantitative real-time PCR (qRT-PCR) was conducted using Perfect-Start™ Green qPCR SuperMix (TransGen Biotech, AQ601) on a CFX96 Touch Real-Time PCR Detection System (Bio-Rad Laboratories, Inc., Hercules, USA). The qRT-PCR primers are listed in Supplementary Table 2. The relative mRNA quantities are expressed as $2^{-\Delta\Delta CT}$.

### RNA-Seq analysis
The RNA integrity was evaluated using a Bioanalyzer 2100 (Agilent, CA, USA), and it was determined to have an RIN number greater than 7.0, which was confirmed through electrophoresis using a denaturing agarose gel. The cDNA libraries were constructed using a SuperScript™ II Reverse Transcriptase Kit (Invitrogen, Cat. No. 18064014), yielding a final cDNA library with an average insert size of $300 \pm 50$ bp. The samples were subjected to paired-end sequencing on an Illumina HiSeq 4000 (LC Sciences) according to the manufacturer's recommended protocol. The mapped reads of each sample were assembled using StringTie (https://ccb.jhu.edu/software/stringtie) with default parameters. Combining transcriptomes from all samples resulted in the reconstruction of a comprehensive transcriptome using Gffcompare (https://github.com/gpertea/gffcompare/). Following the generation of the final transcriptome, StringTie was employed to analyze mRNA expression levels by calculating the FPKM using the following equation: FPKM = total exon fragments/mapped reads (millions) × exon length (kB). Differentially expressed mRNAs were identified using the edgeR package in R according to the following criteria: fold change (FC) ≥ 1.5 or FC ≤ 0.67 and statistical significance ($p < 0.05$). The RNA-Seq data were deposited into the Sequence Read Archive (SRA) database (PRJNA No. PRJNA770508). Gene set enrichment analysis (GSEA) was performed on the differentially expressed genes, which ranked the genes according to their fold changes. Kyoto Encyclopedia of Genes and Genomes (KEGG) pathway enrichment analysis was performed using the clusterProfiler package in R. A pathway was considered significantly enriched if $p$-value < 0.05 and NES > 1.

### Cytoplasmic and nuclear fractionation
Nuclear and cytoplasmic proteins were fractionated using the Nuclear and Cytoplasmic Protein Extraction Kit (Beyotime) according to the manufacturer's instructions. After fractionation, 40 µg of protein was used for western blot analysis of FOXO1 in the cytoplasm and nucleus. β-actin (TransGen Biotech) and Histone H3 (1B1B2) Mouse mAb (Cell Signaling Technology) were used as cytoplasmic and nuclear markers, respectively.

### Transcutaneous measurement of the glomerular filtration rate (GFR)
Mice were anesthetized with isoflurane, and a micro-imaging device (MediBeacon, Germany) was used to detect fluorescence from the skin following intravenous injection of a fluorescent tracer. Before the injection of FITC-sinistrin (7 mg/100 g body weight; MediBeacon, Germany), background fluorescence was recorded for 5 min. During the recording period (-1 h), animals remained conscious and were confined in a customized cage.

Data were recorded using MB-Lab2 software (MediBeacon) and analyzed with MB-studio3 software. The GFR (in µl/min) was calculated based on the fluorescence decay kinetics, which corresponded to the plasma half-life ($t_{1/2}$) of FITC-sinistrin. The GFR was modeled using either (i) a three-compartment model incorporating mouse body weight and empirical conversion factors or (ii) a simpler method based solely on fluorescence decay kinetics.

### ELISA
Plasma or tissue samples were collected from mice subjected to different treatments. Quantitative measurements of the following biomarkers were performed using commercial ELISA kits: FGF4 (Cat. No. FY-EM4425, Feiyue Biotechnology, China), prostate-specific antigen (PSA; Cat. No. PM15725, Pyram, China), carcinoembryonic antigen (CEA; Cat. No. PM18035, Pyram), carbohydrate antigen 125 (CA125; Cat. No. PM18228, Pyram), and alpha-fetoprotein (AFP; Cat. No. PM18628, Pyram). Absorbance at 450 nm was measured using a microplate reader, with subsequent quantification of analyte concentrations derived from standard calibration curves.

### In vivo animal imaging
Alexa Fluor 647 fluorescent dye was mixed with rFGF4 protein and incubated at room temperature for 15 min. The Alexa Fluor 647-labeled rFGF4 protein was then purified using spin columns packed with Bio-Gel P-6 fine resin, following the manufacturer's instructions for the Alexa Fluor 647 Protein Labeling Kit (Cat. No. A30009).

Mice were anesthetized with isoflurane, subjected to dorsal hair removal, and subsequently placed in a thermostatically controlled imager maintained at 37 °C. To minimize background fluorescence interference, mice were fasted for 8 h prior to imaging. For the experiment, mice in the treatment group received an intraperitoneal injection of Alexa Fluor 647-labeled rFGF4, while control animals received an equivalent volume of fluorescently labeled probe. A live animal imaging system (Aniview100, Boluteng Biotechnology, Guangzhou, China) was used to monitor the real-time dynamic distribution of rFGF4 in mice at various time points (0, 15, and 30 min; 1, 2, and 6 h). Six hours after injection, the mice were euthanized. Multiple organs, including the heart, liver, spleen, lungs and kidneys, were collected from each group and imaged using the same equipment. Data analysis was finally performed using AniView Pro software (Version 1.01.01840).

### In vivo drug intervention study
To investigate the effects of current DKD treatments (RAS blockade and SGLT2 inhibitors) on podocyte expression of FGF4 and FGFR1, we employed an STZ-induced DKD mouse model. Following successful model establishment, an 8 week drug intervention was initiated. Mice with DKD were administered either empagliflozin (EMT; HY-15409, MedChemExpress, USA) or losartan potassium (LOS; HY-17512A, MedChemExpress, USA) both at a dose of 10 mg/kg/day via oral gavage. A normal control group and an STZ-induced diabetic non-intervention group received an equal volume of normal saline orally. All mice were euthanized at the conclusion of the 8-week experimental period.

### Statistical analysis
Data analysis was performed using GraphPad Prism 10.1. For experiments involving two groups, statistical comparisons were conducted using the two-tailed Student's t-test. The comparison of data with multiple groups utilized one-way or two-way ANOVA (ordinary or

repeated measure) with post-hoc tests (Tukey or Sidak), as specified in the figure legends. Spearman's correlation test was employed to assess the strength of the association between two variables. A significance level of $p < 0.05$ was considered statistically significant.

## Reporting summary

Further information on research design is available in the Nature Portfolio Reporting Summary linked to this article.

## Data availability

The RNA-seq data in this study have been deposited in the NCBI's Gene Expression Omnibus (GEO) under accession code GSE287026. All other data supporting the findings of this study are included in the manuscript and its Supplementary Information. Source data are provided with this paper.

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

## Acknowledgements
This study was funded by the National Natural Science Foundation of China (82530107, 92357304, and U25A201730 to Z.H.; 81800725 to J.Z.; and U22A20385 to X.L.), the Natural Science Foundation of Zhejiang Province (LDQ24H310001 to Z.H.; MS26H070002 and LQ22H170001 to J.S.), the Key Project from Science Technology Department of Wenzhou (ZY2021022 to Z.H.), the Startup Grant from Oujiang Laboratory (OJQDSP202205 to Z.H.), the Medical and Health Research Project of Zhejiang Province (2024KY1285 to L.J.), the Fundamental Research Funds of Wenzhou Medical University (KYYW202306 to J.Z.), the Zhejiang Province College Students' Science and Technology Innovation Activity Plan (New Talents Plan 2024R413C084), and the Summit Advancement Disciplines of Zhejiang Province (Wenzhou Medical University - Pharmaceutics). We also thank the Scientific Research Center of Wenzhou Medical University for their technical support and the Zhejiang Provincial Key Laboratory of Drug Discovery and Development for Metabolic Diseases.

## Author contributions
J.Z., J.S. and Z.H. conceived the project, designed experiments, analyzed data, and wrote the manuscript. J.Z., S.W., J.L., B.P., M.Z., Q.L., X.C., J.Z., Y.D., S.D., M.Y., J.Z., X.L., J.S. and Z.H. performed experiments, analyzed data, and participated in discussion of the results. L.J., X.W., Y.H., Z.W., and C.Z. provided the human samples and performed the human assay.

## Competing interests
The authors declare no competing interests.
