## [Transparent Peer Review file · Nature Communications]

FGF4-FGFR1 signaling promotes podocyte survival and glomerular function to ameliorate diabetic kidney disease

Corresponding Author: Professor Zhifeng Huang

Version 0:

Reviewer comments:

Reviewer #1

(Remarks to the Author)

This study by Jie Zhou et al. provides evidence for the therapeutic potential of rFGF4 in halting DKD progression by inhibiting oxidative stress to promote podocyte survival and glomerular function via the FGFR1-AMPK-FOXO1 signaling axis. The manuscript is well written and these novel findings will have a meaningful significance to DKD therapy. Some issues that should be addressed to strengthen the conclusion are as follows.

1. Fig1c/1d/1g, the podocin fluorescence is not typical. Podocin is a transmembrane protein of glomerular podocytes, its fluorescence morphology should be similar to that of synaptopodin and nephrin.
2. Fig1c/1d/1g, Extended Data Fig. 1, the FGF4 fluorescence is mostly near circular, which is more likely to be localized in blood cells than in renal intrinsic cells. The same problem occurs with FGFR1 fluorescence in Extended Data Fig. 5.
3. Several previously published scRNA-seq (29980650, 30449713, 33850129) have shown that FGF4 expression is extremely low in podocytes and even in kidneys. Why was FGF4 chosen to study?
4. Whether the effects of intraperitoneal injection of rFGF4 on the renal tubules and other organs of DKD mice could be observed.
5. Part of the original WB is too narrow, and did not show marker, is difficult to determine the molecular weight range. Some figures (e.g., histological images, Western blot quantifications) could benefit from higher resolution and clearer labeling.
6. Fig1h, what is the reason for the collection of shed podocytes in urine of diabetic patients without DKD? In general, the shedding of podocytes is often an indication of early glomerular injury.
7. Fig2f, the glomerular pathology of the WT and Fgf4-PKO groups as shown by HE and Masson staining was abnormal. In addition, Masson staining indicated increased renal tubulointerstitial fibrosis in the Fgf4-KO+STZ group compared to the WT+STZ group. This may be inappropriate.
8. Similarly, the problem of not obvious contrast in glomerular pathology also appeared in Fig3i/ 4JS /6d. Please carefully adjust the image to be more representative and more illustrative.
9. Fig3g/3o, TUNEL-positive cells, that is, apoptotic cells, were all over the visual field of db/db group and STZ group, whether it was inconsistent with the actual renal pathology.
10. Fig4d/4e/6b/6h, The fluorescence specificity of WT1 is poor, it is recommended to change the antibody.
11. Fig4f, why does WB analysis could not detect FGFR1 in glomerular tissue of FGFR1-PKO mice?
12. Results of gene identification in knockout mice should be presented.
13. Recently, a single central administration of FGF4 was reported to have a sustained glucose-controlling effect for more than 7 weeks (37167965). In addition, FGF4 protects the liver from non-alcoholic fatty liver disease by activating the AMPK-Caspase 6 signal axis (35152446). Could you discuss these related mechanisms in more depth?
14. While the study demonstrates that rFGF4 improves podocyte survival and renal function, its long-term effects and safety profile are not addressed. Growth factors like FGF4 could potentially induce undesirable effects such as aberrant cell proliferation or fibrosis.
15. The study focuses on the FGFR1-AMPK-FOXO1 axis but does not explore other potential downstream or interacting pathways, such as mTOR, NF- κ B, or Akt, which are also known to regulate oxidative stress and podocyte apoptosis.
16. The authors should clarify the statistical methods used for comparisons across different experimental groups, particularly for in vivo experiments.
17. There are minor inconsistencies in terminology (e.g., "diabetic nephropathy" vs. "diabetic kidney disease").

Standardizing these terms throughout the manuscript would improve clarity.

18. The clinical sample size is relatively small, which may limit the statistical power of the study. Additionally, the study does not investigate whether FGF4 levels correlate with specific DKD subtypes or progression stages.

Reviewer #2

(Remarks to the Author)

The investigators have provided an extensive series of in vivo and in vitro studies to determine the role and mechanism by which podocyte FGF4 can serve to protect podocyte integrity in response to either type 1 or type 2 diabetes. In general, the studies appear to be well performed and the results are consistent. However, there remain a number of issues.

1) The authors have concentrated on the podocyte but they do not provide adequate information or images about expression of FGF4 or FGFR1 in other nephron structures, and specifically in the tubules. Extended Figure 1 is apparently only showing the glomeruli. There is a previous study (Kidney International 68:2621, 2005) that indicates FGFR1 (as well as FGF1) in tubules as well as glomerular endothelial cells, mesangial cells and parietal epithelial cells in addition to podocytes.

2) Figure 4b needs a podocyte marker

3) Figure 7 Please give more information about the isolated glomeruli. Were the mice treated with FGF4 prior to glomerular isolation. If not, how long were they treated with FGF4 once they were isolated?

4) For Figure 3, the investigators cannot definitively say that the effect of recombinant FGF4 in this type 2 model of diabetes was not due in part or completely to the effect to decrease blood glucose.

5) What is the effect of current therapeutic options for Diabetic Nephropathy (RAS blockade, SGLT2 inhibitors) on podocyte expression of FGF4 and FGFR1?

Reviewer #3

(Remarks to the Author)

To the Authors:

I commend the authors for their work exploring the role of FGF4 in diabetic kidney disease. This is an important area of research, with clear clinical relevance, and the inclusion of human samples and animal models strengthens the study. However, there are several major issues that need to be addressed before the manuscript can be considered for publication.

Major Points:

1. Embryonic Effects of Constitutively Active Cre-Deleter Model: The use of a constitutively active Cre-deleter in this study raises concerns about the potential for developmental phenotypes to confound the observed effects. Given the known role of FGF4 in embryonic development, the authors should clarify whether the effects observed are truly mediated by FGF4 in the adult kidney or whether they might be a consequence of developmental disruptions. An inducible model would be a better approach to avoid embryonic off-target effects and strengthen the manuscript.

2. Mechanism Behind rFGF4 Injections: The authors need to elaborate on the mechanism by which rFGF4 reaches its target organ. As FGF4 is a paracrine factor that requires local signaling with FGFRs and heparan sulfate proteoglycans, it is highly questionable that an intraperitoneal (i.p.) injection of rFGF4 would enter the bloodstream and reach the kidney to act on podocytes at a meaningful concentration. Moreover, since the N-terminus containing the signal peptide is deleted from the rFGF4 construct, the authors need to provide additional data to demonstrate how the protein might be circulating and binding to its targets. This gap in the current literature must be addressed, especially considering that previous studies by the group have failed to clarify this issue.

3. Off-Target Effects of rFGF4: Even assuming rFGF4 is circulating, how can the authors be confident that the observed effects are specifically due to rFGF4 and not off-target effects? Prior studies have demonstrated that FGF4 can affect liver function, and it is plausible that the observed effects could be mediated by changes in circulating inflammatory cytokines or FGF21, which could be synthesized by the liver or adipocytes in response to rFGF4 administration. To address this, the authors should perform additional mechanistic experiments including ELISAs for FGF4 and FGF21, to rule out these possibilities.

4. Discrepant Glucose-Lowering Effects: The manuscript reports inconsistent glucose-lowering effects of rFGF4 in type 1 and type 2 diabetic animal models. Such discrepancies should not be overlooked and should be explained. While addressing these issues might be outside the scope of the current manuscript, at least some discussion or potential explanation is necessary to provide a coherent narrative.

5. FGFR1-cKO Mice and Potential FGF21 Effects: In the FGFR1-cKO mice, it is possible that the described effects are mediated by other factors, such as circulating FGF21. This is particularly relevant since FGF21 can also exert its effects through AMPK and FOXO1 pathways. The authors should consider these additional factors and their potential influence on the observed outcomes.

Minor Points:

Recent advancements in the treatment of kidney disease, particularly in the context of diabetes, should be highlighted in the introduction. The authors should consider updating this section to include recent therapies such as SGLT2 inhibitors, GLP-1 agonists, and non-steroidal mineralocorticoid receptor antagonists (ns-MRAs).

Version 1:

Reviewer comments:

Reviewer #1

(Remarks to the Author)

The authors have addressed all of my issues.

Reviewer #2

(Remarks to the Author)

The authors have successfully responded to my previous concerns and should be congratulated for this very interesting study. I would suggest that you incorporate the new results with the ARB and SGLT2 inhibitor into the text because of the potential clinical relevance.

Reviewer #3

(Remarks to the Author)

Excellent work. All questions and concerns were resolved in my opinion.

Manuscript # NCOMMS-25-07008A

Title: FGF4-FGFR1 signaling promotes podocyte survival and glomerular function to ameliorate diabetic kidney disease

Point-by-point responses to Reviewer's comments

Reviewer #1

This study by Jie Zhou et al. provides evidence for the therapeutic potential of rFGF4 in halting DKD progression by inhibiting oxidative stress to promote podocyte survival and glomerular function via the FGFR1-AMPK-FOXO1 signaling axis. The manuscript is well written and these novel findings will have a meaningful significance to DKD therapy. Some issues that should be addressed to strengthen the conclusion are as follows.

Response: We appreciate the Reviewer's positive feedback on our manuscript. As suggested, we have incorporated substantial additional data to further strengthen the conclusions of our study.

1. Fig1c/1d/1g, the podocin fluorescence is not typical. Podocin is a transmembrane protein of glomerular podocytes, its fluorescence morphology should be similar to that of synaptopodin and nephrin.

Response: We appreciate the Reviewer's valuable comments. In response, we have optimized our experimental approach by employing an enhanced podocin antibody protocol incorporating Tyramide Signal Amplification (TSA) technology (*Theranostics* 2025, PMID: 39776814; *Adv Sci (Weinh)* 2023, PMID: 37802976), which has significantly improved detection sensitivity. The TSA method, as an HRP-based catalytic signal amplification system, offers superior specificity by eliminating potential antibody cross-reactivity issues and overcomes species-

matching constraints between primary and secondary antibodies. This approach provides greater flexibility in experimental design while achieving exponential signal enhancement. The improved staining results obtained through this optimized protocol are presented in **Figure R1**.

To further validate our findings, we have performed additional RNA-Scope re-staining of kidney tissue samples, with the updated experimental results detailed in **Figure R2** (See the revised **Figure 1c, 1d, and 1g**). Moreover, our immunofluorescence co-localization studies of *Fgf4* with WT-1, another podocyte marker, in murine renal tissues also confirmed the specific expression of FGF4 in podocytes (**Figure R3**).

Figure R1. Representative immunofluorescence staining of podocin (red) in renal tissues, with white dashed lines demarcating glomerular boundaries. Nuclei were counterstained with DAPI (blue).

Figure R2. Podocyte-specific expression of FGF4 in DKD models and human renal tissues. **a**, RNA-Scope analysis of *FGF4* mRNA expression (green) in renal tissues from healthy

donors and DKD patients ($n = 10$). **b-e**, *Fgf4* mRNA (green) in podocytes from (**b,c**) *db/db* mice and (**d,e**) STZ-induced diabetic mice ($n = 6$). Podocytes were identified by podocin immunostaining (red), and nuclei were counterstained with DAPI (blue). White dashed lines outline glomerular boundaries. Data are presented as mean \pm SEM. **** $p < 0.0001$ as determined by unpaired two-tailed Student's *t*-test.

Figure R3. Co-localization analysis of *Fgf4* mRNA (green) with podocyte marker WT-1 (red) in renal tissues from *db/db* mice and STZ-induced diabetic mice, with nuclear counterstaining by DAPI (blue). White dashed lines outline glomerular margins. WT-1, Wilms' Tumor 1.

2. *Fig1c/1d/1g*, Extended Data Fig. 1, the FGF4 fluorescence is mostly near circular, which is more likely to be localized in blood cells than in renal intrinsic cells. The same problem occurs with FGFR1 fluorescence in Extended Data Fig. 5.

Response: Thank you for your insightful comments. To address this concern, we have repeated the relevant experiments with optimized protocols. Specifically, during sample preparation of mouse kidney tissues, we performed cardiac PBS perfusion to thoroughly eliminate circulating blood cells, thereby minimizing potential interference. We also applied an autofluorescence quencher treatment to effectively suppress background fluorescence signals originating from residual blood cells and other endogenous sources (**Figure R4**).

To further enhance detection sensitivity and specificity, we conducted a systematic screening of antibody candidates and implemented an optimized Tyramide signal amplification (TSA) protocol. All relevant samples were rigorously reprocessed and the resulting data for both *Fgf4* mRNA and FGFR1 protein expression are now presented in **Figure R5** (See the revised **Figure 1c,d,g** and **Supplementary Fig. 1c-e**) and **Figure R6** (See the revised **Figure 4d,e** and

Supplementary Fig. 8c,d). The experimental protocols for erythrocyte depletion (lines 584-585) and TSA-based immunostaining (lines 709-712) have been provided in the Methods section.

Figure R4. Immunofluorescence staining of renal tissue following perfusion-based removal of circulating blood cells and autofluorescence quenching treatment.

Figure R5. Expression pattern of FGF4 in DKD models and human renal tissues. **a-e**, RNA-
 Scope analysis of *FGF4* mRNA expression (green) in renal tissues from **(a)** healthy donors and
 DKD patients ($n = 10$), **(b,c)** *db/db* mice, and **(d,e)** STZ-induced diabetic mice ($n = 6$). **f-h**,
 co-staining of *Fgf4* RNA-Scope (green) with endothelial marker CD31 (red) or mesangial cell
 marker α -SMA (red) in renal tissues from **(f)** *db/db* mice, **(g)** STZ-induced diabetic mice, and
(h) human DKD specimens. Nuclei were counterstained with DAPI (blue). White dashed lines
 outline glomerular margins. Data are presented as mean \pm SEM. **** $p < 0.0001$ as determined
 by unpaired two-tailed Student's *t*-test.

Figure R6. Podocyte-specific FGFR1 expression in DKD models and human renal tissues. a-c, Immunofluorescence co-staining analysis of FGFR1 expression (green) with (a) podocin (red), (b) α -SMA (red) or (c) CD31 (red) in renal tissues from both STZ-induced diabetic mice (left panels) and human DKD specimens (right panels). Nuclei are counterstained with DAPI (blue).

3. Several previously published scRNA-seq (29980650, 30449713, 33850129) have shown that *FGF4* expression is extremely low in podocytes and even in kidneys. Why was *FGF4* chosen to study?

Response: Thank you for your valuable comments. The primary objective of this investigation was to identify novel modulators within the FGF signaling pathway that contribute to podocyte damage. Through comprehensive PCR-based screening of all FGF family members, we discovered that *FGF4* was the only member consistently downregulated across experimental models. Notably,

despite its low basal expression, FGF4 functions as a classical paracrine signaling molecule whose biological effects are amplified through tyrosine kinase signaling cascades mediated by its receptors (FGFR1/4). Furthermore, our functional studies demonstrated that genetic ablation of *Fgf4* significantly exacerbated podocyte apoptosis and accelerated DKD progression, underscoring its non-redundant role in podocyte homeostasis. These collective findings suggest that FGF4, despite low basal expression, serves as a key protective factor in DKD pathogenesis, highlighting the importance of investigating not only highly expressed genes but also high potent, low-abundance molecular guardians with essential biological functions. In response to these comments, we have incorporated these three key references and made a discussion regarding the "high-potency, low-abundance action of FGF4" and its therapeutic implications in the revised manuscript (**lines 406-409**).

4. Whether the effects of intraperitoneal injection of rFGF4 on the renal tubules and other organs of DKD mice could be observed.

Response: We appreciate the Reviewer for pointing out this important issue. To evaluate the therapeutic effects of rFGF4 on renal tubules, we have reperformed immunohistochemistry and Western blot to assess two well-established tubular injury biomarkers: Neutrophil Gelatinase-Associated Lipocalin (NGAL) and Kidney Injury Molecule-1 (KIM-1), (*Kidney Int* 2020, PMID: 32081304; *Cell Metab* 2021, PMID: 33951465). Our results showed that, compared with the model group, rFGF4 treatment induced a slight reduction of NGAL and KIM-1 level in both DKD models, suggesting that FGF4 may exert a modest protective effect on renal tubules (**Figure R7**).

Furthermore, histological examination of major organs (including heart, liver, spleen, pancreas, and lungs) using H&E staining revealed significant attenuation of hepatic lipid accumulation in *db/db* mice, consistent with our previous reports (*Hepatology*, 2022, PMID: 35152446), However, no pathological alterations were

observed in other major organs following rFGF4 treatment compared to model controls (**Figures R8 and R9**). These results not only confirm the renal protective efficacy of FGF4 in DKD but also highlight its favorable safety profile at the systemic level.

Figure R7. rFGF4 treatment ameliorates renal tubular injury in DKD models. **a,b**, Representative immunohistochemical staining of NGAL in renal tubules from **(a)** *db/db* mice and **(b)** STZ-induced DKD mice, with red arrows referring to the renal tubule. **c,d**, Western blot of KIM-1 protein expression in renal tissues from the two DKD models, with β -actin serving as a loading control. Data are presented as mean \pm s.e.m. * $p < 0.05$, ** $p < 0.001$, **** $p < 0.0001$ as determined by unpaired two-tailed Student's *t*-test **(a, c)** or one-way ANOVA followed by Sidák's multiple comparisons test **(b, d)**. ns, not significant. NGAL, neutrophil gelatinase-associated lipocalin; KIM, Kidney Injury Molecule-1.

Figure R8. Representative H&E staining of major organs (heart, liver, spleen, pancreas, and lungs) following rFGF4 treatment in *db/db* diabetic mice.

Figure R9. Representative H&E staining of major organs (heart, liver, spleen, pancreas, and lungs) following rFGF4 treatment in STZ-induced diabetic mice.

5. Part of the original WB is too narrow, and did not show marker, is difficult to determine the molecular weight range. Some figures (e.g., histological images, Western blot quantifications) could benefit from higher resolution and clearer labeling.

Response: We appreciate the Reviewer's careful examination of our Western blot data. The narrow bands observed in certain blots are mainly due to the presence of multiple target proteins within a single band. The insufficient visibility of molecular weight marker stemmed from the aging imaging system used for Western blot detection.

To address these issues, we have repeated and optimized all relevant Western blots, achieving improved band resolution and marker clarity. The revised data, presented as side-by-side comparisons in **Figures R10-R13**, now demonstrate markedly improved band resolution and enhanced molecular weight marker clarity. All updated Western blot results and corresponding statistical analyses have been incorporated throughout the revised manuscript. We are truly grateful for this constructive feedback, which has strengthened the scientific rigor of our study.

Corresponding to the revised Fig. 2b,h

Figure R10. Re-optimized Western blot results corresponding to the revised Figure 2b,h. The original data (left panel) and newly obtained blots (right panel) are shown for comparison.

Corresponding to the revised Fig. 5e,f,g

Figure R11. Re-optimized Western blot results corresponding to the revised Figure 5e,f,g. The original data (left panel) and newly obtained blots (right panel) are shown for comparison.

Corresponding to the revised Fig. 7d,e

Figure R12. Re-optimized Western blot results corresponding to the revised Figure 7d,e. The original data (left panel) and newly obtained blots (right panel) are shown for comparison.

Corresponding to the revised Supplementary Fig. 5b,d

Figure R13. Re-optimized Western blot results corresponding to the revised Supplementary Fig. 5b,d. The original data (left panel) and newly obtained blots (right panel) are shown for comparison.

6. Fig 1h, what is the reason for the collection of shed podocytes in urine of diabetic patients without DKD? In general, the shedding of podocytes is often an indication of early glomerular injury.

Response: Thank you for your comments. In our study, the term "diabetic patients without DKD" refers to clinically diagnosed diabetic patients who maintain normal renal function indicators. Multiple studies have demonstrated that podocyturia serves as an early marker of glomerular injury, potentially appearing even when renal function and proteinuria remain within normal ranges (*BMC Nephrol* 2025, PMID: 40045253; *J Diabetes Complicat* 2017, PMID: 28161386). In type 2 diabetic patients, podocyturia precedes microalbuminuria and can be detected before reaching the microalbuminuric stage, albeit with relatively low podocyte counts in such cases. Conventional renal biomarkers, such as serum creatinine and proteinuria, typically become detectable after kidney damage has progressed to moderate-to-severe stages (*Lancet Diabetes Endocrinol* 2015, PMID: 25943757). Confirmed DKD patients exhibit significantly more severe podocyte loss. To improve clarity, we have supplemented the definition in the Methods section (**lines 514-516**), specifying that "diabetic patients without DKD" refers to clinically diagnosed diabetic patients who maintain normal renal function indicators.

7. Fig 2f, the glomerular pathology of the WT and Fgf4-PKO groups as shown by HE and Masson staining was abnormal. In addition, Masson staining indicated increased renal tubulointerstitial fibrosis in the Fgf4-KO+STZ group compared to the WT+STZ group. This may be inappropriate.

Response: We sincerely appreciate the Reviewer's comments. Under physiological conditions, both WT and *Fgf4*-PKO mice exhibited comparable glomerular architecture, with no discernible differences observed in H&E or Masson's trichrome staining or in quantitative morphometric assessments. However, following STZ-induced diabetes, while both groups developed glomerular injury, the *Fgf4*-PKO+STZ mice displayed more severe pathological changes, including pronounced mesangial expansion, glomerular basement membrane thickening, and significantly increased collagen deposition, indicating

accelerated fibrosis. In response, we have repeated all relevant histological staining and presented new representative images (**Figure R14**) (See the revised **Figure 2f**).

Regarding the observed exacerbation of tubulointerstitial fibrosis in *Fgf4*-PKO+STZ mice, we propose this phenotype results from pathogenic cross-talk between injured podocytes and tubulointerstitial cells. This aligns with emerging evidence that injured podocytes drive tubulointerstitial pathology through multiple mechanisms, such as paracrine secretion of profibrotic mediators or exosome-mediated transfer of regulatory molecules (*Kidney Int* 2024, PMID: 37774921; *Commun Biol* 2022, PMID: 35869269; *Clin Sci (Lond)* 2020, PMID: 32478397). These mechanisms position podocytes as central orchestrators of diabetic nephropathy progression, bridging glomerular and tubulointerstitial damage.

Figure R14. The glomerular pathology in WT and *Fgf4*-PKO mice. Representative images and quantification analysis of renal tissue sections stained with H&E and Masson's trichrome.

8. Similarly, the problem of not obvious contrast in glomerular pathology also appeared in Fig3i/ 4jS /6d. Please carefully adjust the image to be more representative and more illustrative.

Response: Thank you for your detailed review and feedback. Regarding the issue of insufficient contrast in the H&E staining results shown in Figures 3i, 4j, and 6d of the glomerular pathology images, we have carefully reviewed the original data

and made corresponding adjustments (Figures R15-R17) (See the revised Figures 3d, 3l, 4j, and 6d). We believe the revised results now provide a more representative and accurate assessment of glomerular pathology.

Corresponding to the revised Fig. 3d,l

Figure R15. Therapeutic effects of rFGF4 on glomerular pathology in DKD model mice. a,b, Representative H&E and Masson's trichrome-stained renal sections from (a) *db/db* mice and (b) STZ-induced diabetic mice following rFGF4 treatment.

Corresponding to the revised Fig. 4j

Figure R16. H&E staining of renal tissue sections from WT and *Fgfr1*-PKO diabetic mice treated with or without rFGF4.

Corresponding to the revised Fig. 6d

Figure R17. H&E staining of renal tissue sections from WT and *Ampk*-PKO diabetic mice treated with or without rFGF4.

9. Fig 3g/3o, TUNEL-positive cells, that is, apoptotic cells, were all over the visual field of *db/db* group and STZ group, whether it was inconsistent with the actual renal pathology.

Response: We sincerely appreciate the Reviewer's insightful observations regarding apoptosis detection. The initial 40× objective magnification images presented in our manuscript were selected to demonstrate representative fields with clearly detectable apoptotic signals, which might create an impression of more widespread positivity than actually observed. To provide a more accurate representation of the apoptotic cell distribution, we have meticulously repeated all relevant staining and provide 20× overview images that better reflect the actual focal distribution of TUNEL-positive cells across entire tissue sections, along with corresponding 40× magnified views in **Figure R18** (See the revised **Figure 3g,o**).

Our revised analysis demonstrated while both *db/db* and STZ diabetic groups exhibited significantly increased TUNEL-positive cell counts compared to control groups, but these apoptotic cells were not distributed throughout the entire visual field. This observed sensitivity aligns with established performance of the Promega TUNEL assay kit (Cat#G3250) employed in our study, which has been validated in previous studies (*Cell Death Dis* 2025, PMID: 39979265; *J Cell Mol Med* 2019, PMID: 31286669) for its high sensitivity and specificity in detecting apoptotic cells.

Figure R18. Apoptotic cell detection in DKD mouse kidneys. a-d, Representative TUNEL staining (green) of renal sections from (a,b) STZ-induced and (c,d) *db/db* diabetic mice, with nuclei counterstained with DAPI (blue). Data presented as mean \pm SEM. **** $p < 0.0001$ as determined by Ordinary one-way ANOVA followed by Dunnett's multiple comparisons test (b) or unpaired two-tailed Student's *t*-test (d).

10. Fig 4d/4e/6b/6h, The fluorescence specificity of WT1 is poor, it is recommended to change the antibody.

Response: Thank you for your comments. As suggested, we have replaced the WT-1 antibody with a Podocin antibody and repeated all relevant experiments. The updated results obtained with this improved antibody are now presented in **Figure R19** (See the revised **Figure 4d,e**) and **Figure R20** (See the revised **Figure 6b,h**).

Figure R19. Podocyte-specific expression of FGFR1 in human and mouse renal tissues. **a,b**, Dual immunofluorescence staining showing co-localization of FGFR1 (green) with podocyte (red) in glomeruli from **(a)** STZ-induced diabetic mice and **(b)** human DKD specimens. Nuclei are counterstained with DAPI (blue) in both panels.

Figure R20. AMPK and podocin expression in Podocyte-specific *Ampk* knockout

diabetic mice. a, Immunofluorescence staining of glomeruli showing co-localization of AMPK (green) with the podocyte (red). **b**, Representative images and quantitative analysis of podocyte (green) staining in renal sections from WT and *Ampk*-PKO diabetic mice treated with or without rFGF4 ($n = 6$). Nuclei were counterstained with DAPI (blue). Data are presented as mean \pm s.e.m. $**p < 0.01$, $***p < 0.001$ as determined by Ordinary one-way ANOVA. ns, no significance.

11. Fig 4f, why does WB analysis could not detect FGFR1 in glomerular tissue of FGFR1-PKO mice?

Response: Thank you for your comments. Our original findings demonstrated that FGFR1 is expressed predominantly in glomerular podocytes (**Figure R19**). To further validate this observation, we have performed additional immunofluorescence staining of renal tissues from both WT and *Fgfr1*-PKO mice. These additional experiments clearly revealed prominent FGFR1 localization in WT podocytes, while showing nearly undetectable expression in podocytes of knockout mice (**Figure R21**) (See also **Supplementary Fig. 9b** in the revised manuscript), which is consistent with our Western blot results.

Figure R21. Efficient knockout of *Fgfr1* in podocytes following glomerulus-specific *Fgfr1* deletion. Immunofluorescence staining demonstrates the near-complete absence of FGFR1 (green) in glomeruli of conditional knockout mice compared to WT controls, confirming successful podocyte-specific *Fgfr1* ablation. Nuclei were counterstained with DAPI (blue).

12. Results of gene identification in knockout mice should be presented.

Response: We thank the reviewer for highlighting this important point. As suggested, we have now included the genotyping results for all four knockout

mouse models in **Figure R22**. These new data have been added to **Supplementary Fig. 3a, 9a, and 11a,b** in the revised manuscript.

Figure R22. Genotyping of podocyte-specific knockout mice. PCR-based gene identification confirms successful knockout of *Fgf4* (a), *Fgfr1* (b), *Ampka1/α2* (c), and *Foxo1* (d) in podocyte-specific conditional knockout mice. Representative gel electrophoresis images show wild-type (WT), floxed (fl/fl), and knockout (KO) genotypes for each target gene. bp, base pair.

13. Recently, a single central administration of FGF4 was reported to have a sustained glucose-controlling effect for more than 7 weeks (37167965). In addition, FGF4 protects the liver from non-alcoholic fatty liver disease by activating the AMPK-Caspase 6 signal axis (35152446). Could you discuss these related mechanisms in more depth?

Response: Thank you for your insightful comments. Indeed, our previous research has established that FGF4 plays a crucial role in modulating central glucose levels and ameliorating non-alcoholic fatty liver disease (NAFLD). Mechanistic studies suggested that the influence of FGF4 on both central glucose regulation and NAFLD mitigation were mediated through FGFR-dependent AMPK signaling in target cells.

Although these tissue-specific actions involve distinct FGFR subtypes, these two studies - along with our current findings - consistently identify AMPK as a central molecular node in FGF4's biological network. This represents a significant discovery, underscoring FGF4 as a potent AMPK activator with substantial therapeutic potential for glucose and lipid metabolism dysregulation, as well as related pathologies. We have expanded this discussion (**lines 446-450**) in the revised manuscript.

14. While the study demonstrates that rFGF4 improves podocyte survival and renal function, its long-term effects and safety profile are not addressed. Growth factors like FGF4 could potentially induce undesirable effects such as aberrant cell proliferation or fibrosis.

Response: We thank the reviewer for highlighting this important point. First, the rFGF4 used in this study carries an engineered N-terminal truncation (residues Ala67 to Leu206), which has been demonstrated to exhibit diminished mitogenic activity (*Nat Commun* 2021, PMID: 34907199; *Cell Metab* 2023, PMID: 37167965). In direct response to this comment, we performed safety assessments in both *db/db* and STZ-induced DKD models following chronic rFGF4 administration for two months. Our results revealed that serum tumor markers (AFP, PSA, CA125, and CEA) remained within normal physiological ranges without significant treatment-related aberrations, and no detectable increases in Ki67-positive foci or cell cycle gene expression were observed (**Figures R23 and R24**) (See also **Supplementary Fig. 6** in the revised manuscript). Furthermore, histopathological examinations demonstrated no morphologic abnormalities in major organs beyond the expected therapeutic renal improvements (**Figures R8 and R9**). In addition, we found that rFGF4 has a certain improvement effect on renal fibrosis (**Supplementary Fig. 5**). Collectively, these safety assessments, spanning molecular, biochemical, and histological analyses, provide compelling evidence supporting rFGF4's favorable safety profile for DKD therapy.

Figure R23. Safety assessment of chronic rFGF4 administration in *db/db* mice. **a**, Changes of serum levels of tumor markers. **b**, Quantitative analysis of proliferation-related markers gene expression (*Ki67*, *Cdk4*) by RT-qPCR. **c**, Representative immunohistochemical staining of Ki67 in major organs. $n = 6$ mice per group. Data are presented as mean \pm s.e.m. ns, not significant as determined by unpaired two-tailed Student's *t*-test. AFP, Alpha-Fetoprotein; PSA, Prostate-Specific Antigen; CA125, Carbohydrate Antigen 125; CEA, Carcinoembryonic Antigen.

Figure R24. Safety assessment of chronic rFGF4 administration in STZ-induced diabetic mice. **a**, Changes of serum levels of tumor markers. **b**, Quantitative analysis of proliferation-related markers gene expression (*Ki67*, *Cdk4*) by RT-qPCR. **c**, Representative immunohistochemical staining of Ki67 in major organs. $n = 6$ mice per group. Data are

presented as mean \pm s.e.m. ns, not significant as determined by Ordinary one-way ANOVA followed by Tukey's multiple comparisons test. AFP, Alpha-Fetoprotein; PSA, Prostate-Specific Antigen; CA125, Carbohydrate Antigen 125; CEA, Carcinoembryonic Antigen.

15. The study focuses on the FGFR1-AMPK-FOXO1 axis but does not explore other potential downstream or interacting pathways, such as mTOR, NF- κ B, or Akt, which are also known to regulate oxidative stress and podocyte apoptosis.

Response: We sincerely appreciate your valuable suggestions regarding additional signaling pathways. Our initial sequencing data revealed predominant enrichment of the AMPK-FOXO1 signaling pathway, which was subsequently validated through comprehensive experiments employing FGFR1, AMPK, and FOXO1 knockout mouse models as well as siRNA knockdown approaches, conclusively demonstrating that FGF4 exerts its effects via the FGFR1-AMPK-FOXO1 axis. In direct response to this comment, we performed systematic evaluations of mTOR, NF- κ B, and Akt signaling in our STZ-induced DKD models. The results revealed that rFGF4 treatment had minimal effects on the AKT and mTOR pathways but suppressed NF- κ B activation (**Figure R25**).

As the reviewer rightly notes, NF- κ B activation is a well-established driver of renal injury through sustained chronic inflammation and oxidative stress (*Kidney Int* 2024, PMID: 38789037). Notably, AMPK, a central cellular energy sensor, has been shown to suppress NF- κ B signaling through multiple molecular mechanisms (*Circ Res* 2016, PMID: 27439892; *Metabolism* 2018, PMID: 29526538; *Redox Biol* 2017, PMID: 28285192). Our findings suggest that FGF4 may play a regulatory role in inflammatory signaling during DKD progression. Our future studies will prioritize elucidating the anti-inflammatory mechanisms of FGF4 in this context.

Figure R25. Evaluation of FGF4's effects on other potential signaling pathways. Western blot analysis of phosphorylated- (p-) AKT, p-mTOR, and p-NF- κ B p65 in renal tissues from rFGF4-treated STZ-diabetic mice, with corresponding total protein as controls ($n = 4$). Data are presented as mean \pm s.e.m. * $p < 0.05$, **** $p < 0.0001$ as determined by Ordinary one-way ANOVA followed by Tukey's multiple comparisons test. ns, not significant.

16. *The authors should clarify the statistical methods used for comparisons across different experimental groups, particularly for in vivo experiments.*

Response: Thank you for your comments. We have meticulously detailed the statistical methodologies employed for inter-group comparisons in each figure legend's concluding section. As comprehensively described in our Methods, between-group analyses were performed using two-tailed Student's t-tests for dual-group comparisons, while multi-group datasets were analyzed via either one-way or two-way ANOVA (employing either ordinary or repeated measures design, as appropriate) followed by specified post-hoc testing (Tukey's or Sidak's correction), with all specific test applications explicitly indicated in respective figure legends.

17. *There are minor inconsistencies in terminology (e.g., "diabetic nephropathy" vs. "diabetic kidney disease"). Standardizing these terms throughout the manuscript would improve clarity.*

Response: We sincerely appreciate your suggestion regarding terminology

standardization. To improve clarity, we have systematically replaced all instances of "diabetic nephropathy" with "diabetic kidney disease" throughout the revised manuscript.

18. The clinical sample size is relatively small, which may limit the statistical power of the study. Additionally, the study does not investigate whether FGF4 levels correlate with specific DKD subtypes or progression stages.

Response: In response to the reviewer's valuable suggestion, we have expanded the clinical cohort by increasing the number of urine samples from 16 to 44 and renal tissue samples from 15 to 22, resulting in a total of 66 clinical specimens, with the comprehensive clinical details now presented in the revised **Supplementary Table 1**. The additional samples were subjected to FGF4 immunofluorescence staining, and the updated quantitative analysis confirms a significant reduction of FGF4 protein levels in podocytes from DKD patients compared to diabetic controls without nephropathy (**Figure R26a**) (See the revised **Figure 1h**). Furthermore, we performed stratified correlation analysis between urinary albumin-to-creatinine ratio (UACR) and podocyte FGF4 immunofluorescence intensity across stages of DKD progression (Phase II-IV), which reveals a stage-dependent correlation pattern during disease progression (**Figure R26b**) (See the revised **Figure 1i**).

Figure R26. FGF4 expression is negatively correlated with the progression of DKD. a, Quantification of FGF4 protein levels in primary human podocytes isolated from diabetic patients with ($n = 29$) and without ($n = 15$) nephropathy. **b,** Negative correlation between UACR and FGF4 immunofluorescence intensity in DKD patients at different progression stages

($n = 29$). Data are presented as mean \pm s.e.m. **** $p < 0.0001$ as determined by unpaired two-tailed Student's t -test. UACR, urinary albumin-to-creatinine ratio.

Reviewer #2

The investigators have provided an extensive series of in vivo and in vitro studies to determine the role and mechanism by which podocyte FGF4 can serve to protect podocyte integrity in response to either type 1 or type 2 diabetes. In general, the studies appear to be well performed and the results are consistent. However, there remain a number of issues.

Response: We appreciate the Reviewer's positive and thoughtful evaluation of our work. In response to the comments, we have performed a substantial amount of additional data to further strengthen the conclusions of our study.

1. The authors have concentrated on the podocyte but they do not provide adequate information or images about expression of FGF4 or FGFR1 in other nephron structures, and specifically in the tubules. Extended Figure 1 is apparently only showing the glomeruli. There is a previous study (Kidney International 68:2621, 2005) that indicates FGFR1 (as well as FGF1) in tubules as well as glomerular endothelial cells, mesangial cells and parietal epithelial cells in addition to podocytes.

Response: We appreciate the Reviewer's insightful reference to prior work (Kidney Int 68:2621, 2005) that documented FGFR1 expression across multiple renal cell populations. Notably, this earlier study confirmed that podocytes serve as the primary cellular site of FGFR1 expression within glomeruli under physiological conditions - a finding that aligns well with our current observations. To further clarify the expression pattern of FGF4, we performed double-labeling experiments combining *Fgf4* RNA-scope in situ hybridization with immunostaining for the specific tubular marker AQP-1. The results confirmed that *Fgf4* is predominantly localized to glomeruli, with relatively low expression levels

detected in tubular compartments (**Figure R27**). For FGFR1 characterization, we employed the same dual-labeling approach combined with AQP-1 for verification. Results indicated that FGFR1 is primarily expressed in glomeruli, which is consistent with our initial findings (**Figure R28**) (See also **Supplementary 8b**).

Figure R27. Expression pattern of FGF4 in DKD models and human renal tissues. a, Immunofluorescence co-localization analysis of *Fgf4* mRNA (green) with tubular marker AQP-1 (red) in renal tissues. White dashed lines delineate podocyte boundaries.

Figure R28. Expression pattern of FGFR1 in normal C57BL/6 mice. Immunofluorescence co-localization analysis depicting FGFR1 (green) and tubular marker AQP-1 (red) in renal tissues. White dashed lines outline podocyte boundaries.

2. *Figure 4b needs a podocyte marker.*

Response: We thank the reviewer for this suggestion. As requested, we have now performed co-staining experiments for FGFR1 and podocin (a podocyte marker), with results demonstrating that FGFR1 localization is primarily confined to podocytes (**Figure R29**) (See also **Supplementary Fig. 8a** in the revised manuscript), further supporting our previous findings regarding FGFR1's podocyte-specific expression pattern.

Figure R29. Immunofluorescence analysis of FGFR1 expression in renal tissues. Normal C57BL/6 mouse renal sections showing FGFR1 localization. White dashed lines delineate glomerular boundaries.

3. *Figure 7 Please give more information about the isolated glomeruli. Were the mice treated with FGF4 prior to glomerular isolation. If not, how long were they treated with FGF4 once they were isolated?*

Response: We appreciate the Reviewer's comments. For human sample experiments, glomeruli were meticulously isolated from clinically discarded renal tissues obtained from nephrectomy specimens of patients with renal carcinoma. Samples were then subjected to high glucose (33 mM) with or without rFGF4 treatment (200 ng/ml) for 36 hours. In murine experiments, both *Fgf4*-PKO and *Fgfr1*-PKO mice received chronic rFGF4 treatment, after which glomeruli were isolated and fixed in 4% paraformaldehyde (PFA) for subsequent immunofluorescence analyses, with the exception of *Foxo1*-PKO mice. The glomerular isolation and treatment protocols for *Foxo1*-PKO mice were consistent with those used for human samples. Detailed protocols have been revised in the Methods section (**lines 484-511**).

4. *For Figure 3, the investigators cannot definitively say that the effect of recombinant FGF4 in this type 2 model of diabetes was not due in part or completely to the effect to decrease blood glucose.*

Response: Thank you for your comments. Our study demonstrates that FGF4 exerts renoprotective effects in DKD models of both type 1 and type 2 diabetes.

However, we observed differential effects of FGF4 on glycemic control. It significantly reduced blood glucose levels in type 2 diabetes but no such effect was seen in type 1 diabetes. Therefore, we speculated that the impact of FGF4 on type 1 diabetes is independent of blood glucose reduction. However, for type 2 diabetes, we cannot definitively conclude that the benefits are entirely independent of glucose-lowering effects. In response to your insightful suggestions, we have carefully revised the manuscript by removing all claims regarding "independently of its hypoglycemic effects". We are grateful for your expert guidance, which has significantly enhanced the scientific accuracy and rigor of our manuscript.

5. What is the effect of current therapeutic options for Diabetic Nephropathy (RAS blockade, SGLT2 inhibitors) on podocyte expression of FGF4 and FGFR1?

Response: This is an excellent question. Currently, there are no published reports indicating that RAS blockers or SGLT2 inhibitors can significantly regulate the expression of FGF4/FGFR1 in podocytes. As suggested by the reviewer, we established an STZ-induced DKD mouse model and administered either RAS blockade Losartan potassium (LOS, 10 mg/kg, Cat#HY-17512A, MedChemExpress) or the SGLT2 inhibitor Empagliflozin (EMP, 10 mg/kg, Cat#HY-15409, MedChemExpress) via oral gavage once daily. After two months of treatment, the mice were euthanized, and renal tissues were collected for Western blot analyses.

Our results showed that blood glucose levels were significantly reduced in the EMP-treated group, whereas LOS treatment had no significant effect on blood glucose. Compared with the STZ model group, the LOS-treated group exhibited upregulation of both FGF4 protein and gene expression in glomerular tissue, suggesting a potential mechanism of LOS in promoting the expression of FGF4, thereby ameliorating podocyte injury. In contrast, EMP treatment did not alter renal FGF4 expression relative to STZ controls. In addition, neither LOS or EMP treatment significantly affected FGFR1 expression levels (**Figure R30**).

Figure R30. Therapeutic effects of RAS blockade and SGLT2 inhibitor on podocyte FGF4 and FGFR1 expression. **a**, Blood glucose levels across treatment groups. **b,c**, qRT-PCR analysis of **(b)** *Fgf4* and **(c)** *Fgf1* mRNA levels in isolated glomeruli ($n = 6$). **d,e**, Western blot analysis of **(d)** FGF4 and **(e)** FGFR1 protein expression in glomerular lysates, with β -actin serving as loading control ($n = 4$). Data are presented as mean \pm s.e.m. * $p < 0.05$, *** $p < 0.001$, **** $p < 0.0001$ as determined by Ordinary one-way ANOVA followed by Tukey's multiple comparisons test. ns, not significant. LOS, Losartan potassium; EMP, Empagliflozin.

Reviewer #3:

I commend the authors for their work exploring the role of FGF4 in diabetic kidney disease. This is an important area of research, with clear clinical relevance, and the inclusion of human samples and animal models strengthens the study. However, there are several major issues that need to be addressed before the manuscript can be considered for publication.

Response: We sincerely appreciate the Reviewer's positive reception of our manuscript and the constructive suggestions provided to enhance it quality.

Major Points:

1. *Embryonic Effects of Constitutively Active Cre-Deleter Model: The use of a*

constitutively active Cre-deleter in this study raises concerns about the potential for developmental phenotypes to confound the observed effects. Given the known role of FGF4 in embryonic development, the authors should clarify whether the effects observed are truly mediated by FGF4 in the adult kidney or whether they might be a consequence of developmental disruptions. An inducible model would be a better approach to avoid embryonic off-target effects and strengthen the manuscript.

Response: We appreciate your insightful and expert comments. This concern you raised is indeed of significant scientific importance, particularly for a multifunctional factor like FGF4 that plays critical roles in embryonic development. In response, we have systematically reviewed our original data and conducted additional experiments to provide the following evidence supporting our conclusion that the observed phenotypes primarily result from *Fgf4* deficiency in adult kidneys rather than developmental defects.

In nondiabetic adult *Fgf4*-PKO mice, renal function (UACR, BUN, GFR; Fig. 2c-e) and histopathological examination (H&E, Masson staining; Fig. 2f) revealed no significant differences compared to WT controls, indicating that conditional *Fgf4* deletion in adulthood does not alter renal structure or function. Complementing these observations, histological analysis of developing kidneys from both neonatal (7-day-old) and young (8-week-old) WT and *Fgf4*-PKO mice showed preserved renal morphology, with normal nephron numbers, glomerular morphology, collecting duct branching, and interstitial organization (**Figure R31**).

We fully concur with the Reviewer's suggestion that utilizing an inducible Cre model (e.g., NPHS2-CreERT2) for adult-stage podocyte-specific *Fgf4* deletion would represent the optimal approach to validate our findings. However, the establishment of such an inducible Cre model is a time-intensive process. As an alternative, we generated an inducible knockout model by administering AAV-Cre to induce site-specific deletion in the kidneys of adult *Fgf4*^{flox/flox} mice (*Fgf4*-RKO; **Figure R32a,b**). Blood glucose levels were comparable between *Fgf4*-RKO mice and WT mice under both non-diabetic (-STZ) and diabetic (+STZ) conditions

(**Figure R32c**). However, diabetic *Fgf4*-RKO mice exhibited significantly more severe renal dysfunction, as evidenced by elevated UACR and BUN (**Figure R32d**). Histopathological analyses using H&E, PAS, and Masson's trichrome staining revealed minimal changes in non-diabetic *Fgf4*-RKO mice but markedly exacerbated diabetic nephropathy features (glomerular mesangial expansion, glycogen deposition, collagen accumulation) in diabetic *Fgf4*-RKO mice (**Figure R32e**). Ultrastructural analysis revealed thickening of the glomerular basement membrane (GBM), foot process fusion, and even foot process disappearance in both WT and *Fgf4*-RKO diabetic mice, with the latter exhibiting more severe pathological alterations (**Figure R32f**). Furthermore, diabetic mice displayed significant downregulation of Nephryn and Podocin expression compared to non-diabetic controls, with *Fgf4*-RKO diabetic mice showing even greater reduction than their diabetic WT controls (**Figure R32g**).

Collectively, findings from both constitutive and inducible knockout models strongly support our conclusion that the adult renal phenotypes observed in this study primarily reflect the direct consequences of FGF4 signaling deficiency in adult kidney cells, rather than secondary effects of developmental disruptions.

Figure R31. Renal histology in developing *Fgf4*-deficient mice. Representative H&E-stained renal sections from WT and *Fgf4*-RKO mice at neonatal (7-day-old) and juvenile (8-week-old) stages show comparable renal architecture.

Figure R32. Renal *Fgf4* deletion exacerbates diabetes-induced renal injury. **a**, Schematic of renal *Fgf4* knockout (*Fgf4*-RKO) via intrarenal AAV2/8-Cre delivery. **b**, Western blot analysis of renal FGF4 expression. **c**, Blood glucose levels in WT and *Fgf4*-RKO mice under nondiabetic or diabetic conditions ($n = 8$). **d**, Alterations in UACR and BUN in renal tissues ($n = 6$). **e**, Representative H&E, PAS, and Masson's trichrome staining with quantitative histopathology ($n = 8$). **f**, TEM and quantification of renal tissues from the indicated groups ($n = 12$). Yellow arrows refer to basement membranes. **g**, Western blot and quantification of podocyte-specific markers (Nephrin and Podocin) in total renal lysates from the indicated groups ($n = 6$). β -actin served as a loading control. Data are presented as mean \pm SEM. * $p < 0.05$, ** $p < 0.01$ as determined by ordinary two-way ANOVA followed by Sidak's multiple comparisons; ns, not significant. BUN, Blood urea nitrogen; TEM, Transmission electron microscopy.

2. Mechanism Behind rFGF4 Injections: The authors need to elaborate on the mechanism by which rFGF4 reaches its target organ. As FGF4 is a paracrine factor

that requires local signaling with FGFRs and heparan sulfate proteoglycans, it is highly questionable that an intraperitoneal (i.p.) injection of rFGF4 would enter the bloodstream and reach the kidney to act on podocytes at a meaningful concentration. Moreover, since the N-terminus containing the signal peptide is deleted from the rFGF4 construct, the authors need to provide additional data to demonstrate how the protein might be circulating and binding to its targets. This gap in the current literature must be addressed, especially considering that previous studies by the group have failed to clarify this issue.

Response: We thank the Reviewer for this insightful comment. To address these concerns, we performed comprehensive pharmacokinetic and tissue distribution analyses. Our ELISA-based analyses revealed that rFGF4 could be effectively absorbed into circulation following intraperitoneal administration, with plasma concentrations peaking at 12 hours post-injection and returning to baseline by 48 hours (**Figure R33a**). Notably, tissue distribution profiling showed preferential renal accumulation of rFGF4, with maximal kidney concentrations detected at 6 hours post-injection (**Figure R33b**). To further corroborate these findings, we conducted fluorescence tracing experiments using FITC-labeled rFGF4, which confirmed significant renal accumulation following intraperitoneal administration (**Figure R34**). Collectively, these results provide compelling evidence that rFGF4 can effectively reach kidney tissues via this administration route, supporting its therapeutic applicability for kidney diseases. All supporting data have been included in **Supplementary Fig. 7**.

Regarding the concern about the signal peptide deletion, we confirm that the absence of the signal peptide does not impair receptor binding or downstream signaling activation. Structural and functional studies demonstrate that FGF4's functional domains reside in its core β -trefoil region (residues 79–170), while the N-terminal signal peptide (residues 1–30) solely directs endoplasmic reticulum-dependent secretion without affecting the mature protein's receptor-binding capacity (*Mol Cell Biol* 2001, PMID: 11486033; *Pharmacol Res* 2020, PMID:

31770593). Notably, endogenous FGF4 is naturally processed to remove its signal peptide prior to functional activity. As our study utilized exogenous rFGF4 administration, the absence of the signal peptide has no impact on its therapeutic efficacy.

Figure R33. Temporal expression profiles of rFGF4 following intraperitoneal administration. **a**, Serum concentration-time profile of rFGF4 measured by ELISA. **b**, Time-dependent accumulation of rFGF4 in renal tissues. *n* = 3/time point.

Figure R34. Tissue distribution of fluorescently labeled FGF4. **a**, *In vivo* fluorescence

imaging of B6 mice at different time points after intraperitoneal injection of Alexa Fluor 647-FGF4, with unconjugated Alexa Fluor 647 control shown for comparison. **b**, *Ex vivo* fluorescence imaging of major organs harvested 6 hours post-administration, illustrating the biodistribution pattern of Alexa Fluor 647-FGF4.

3. Off-Target Effects of rFGF4: Even assuming rFGF4 is circulating, how can the authors be confident that the observed effects are specifically due to rFGF4 and not off-target effects? Prior studies have demonstrated that FGF4 can affect liver function, and it is plausible that the observed effects could be mediated by changes in circulating inflammatory cytokines or FGF21, which could be synthesized by the liver or adipocytes in response to rFGF4 administration. To address this, the authors should perform additional mechanistic experiments including ELISAs for FGF4 and FGF21, to rule out these possibilities.

Response: We sincerely appreciate the Reviewer's comment regarding the potential off-target effects of rFGF4. In direct response to this concern, we conducted pharmacokinetic and mechanistic studies in STZ-induced diabetic mice. Our ELISA analyses of serum samples demonstrated time-dependent changes in FGF4 levels following rFGF4 administration, while both FGF21 and the inflammatory cytokine IL-6 remained at baseline concentrations throughout the observation period. Additionally, parallel examinations of hepatic FGF21 expression at multiple time points confirmed the absence of significant alterations, consistent with the serum findings (**Figure R35**). These results indicate that the renoprotective effects of rFGF4 are mediated through direct FGF4-FGFR1 signaling rather than secondary mechanisms involving FGF21 pathway activation or systemic inflammatory responses.

Figure R35. Temporal expression profiles of key factors following rFGF4 treatment in STZ-induced diabetic mice. a-c, Serum levels of (a) FGF4, (b) IL-6, and (c) FGF21 were quantified by ELISA at indicated time points post-treatment. d, Renal FGF21 expression dynamics were similarly assessed in kidney tissue lysates. Data are presented as mean \pm SEM.

4. *Discrepant Glucose-Lowering Effects: The manuscript reports inconsistent glucose-lowering effects of rFGF4 in type 1 and type 2 diabetic animal models. Such discrepancies should not be overlooked and should be explained. While addressing these issues might be outside the scope of the current manuscript, at least some discussion or potential explanation is necessary to provide a coherent narrative.*

Response: We appreciate the Reviewer's insightful comments regarding the glucose-lowering effects of FGF4. As demonstrated in our previous study (*Nat Commun* 2021, PMID: 34907199), long-term administration of rFGF4 effectively maintains glucose homeostasis in type 2 diabetes (T2D) mice through dual mechanisms: ameliorating insulin resistance in skeletal muscle and suppressing adipose tissue macrophage inflammation. However, the mechanism of metabolic derangement in type 1 diabetes (T1D) is fundamentally distinct from that in T2D

mice, characterized by "absolute insulin deficiency" rather than "insulin signaling dysfunction". For T1D treatment, therapeutic approaches that neither directly supplement insulin nor restore β -cell function are unlikely to correct blood glucose. We believe that the lack of hypoglycemic effect of FGF4 in T1D mice may be due to its inability to substitute for insulin across glucose-regulating tissues (e.g., muscle, liver) and potential suppression of signaling pathways under an insulin-deficient metabolic state. As evidenced by our previous report (*Nat Commun* 2021, PMID: 34907199), rFGF4 administration did not promote an increase in insulin levels. Moreover, incubation with rFGF4 did not induce AKT phosphorylation in L6 myocytes with insulin-deficiency. We have incorporated this discussion in the revised manuscript (**lines 422-425**).

5. FGFR1-cKO Mice and Potential FGF21 Effects: In the FGFR1-cKO mice, it is possible that the described effects are mediated by other factors, such as circulating FGF21. This is particularly relevant since FGF21 can also exert its effects through AMPK and FOXO1 pathways. The authors should consider these additional factors and their potential influence on the observed outcomes.

Response: Thank you for your comments. As shown in our response in **Figure R35**, rFGF4 administration did not alter serum or hepatic FGF21 levels in diabetic mice. rFGF4 administration showed no detectable effect on FGF21 expression levels. To further investigate potential FGFR1-dependent regulation of circulating FGF21, we performed comparative analyses of serum and hepatic FGF21 levels between diabetic *Fgfr1*-WT and *Fgfr1*-cKO mice. Our data revealed no significant differences in either circulating (serum) or tissue (liver) FGF21 expression between these genotypes (**Figure R36**). These comprehensive results provide compelling evidence that the renoprotective effects of FGF4 in DKD are mediated through direct actions on glomerular cells rather than secondary mechanisms involving FGF21 pathway modulation.

Figure R36. Circulating and hepatic FGF21 levels in *Fgfr1*-cko diabetic mice. Serum FGF21 concentrations and hepatic FGF21 expression levels in STZ-induced diabetic *Fgfr1*-WT and *Fgfr1*-PKO mice, as quantified by ELISA ($n = 5$ per group). Data are presented as mean \pm SEM. ns, not significance.

Minor Points:

Recent advancements in the treatment of kidney disease, particularly in the context of diabetes, should be highlighted in the introduction. The authors should consider updating this section to include recent therapies such as SGLT2 inhibitors, GLP-1 agonists, and non-steroidal mineralocorticoid receptor antagonists (ns-MRAs).

Response: Thank you for your comments. We have carefully reviewed the literature and incorporated the most recent advancements in diabetic kidney disease treatment strategies in the introduction section (**lines 61-65**) of our revised manuscript. We are grateful for your expert guidance, which has helped strengthen the overall quality and impact of our work.

Manuscript # NCOMMS-25-07008B

Title: FGF4-FGFR1 signaling promotes podocyte survival and glomerular function to ameliorate diabetic kidney disease

Point-by-point responses to Reviewer's comments

Reviewer #1

The authors have addressed all of my issues.

Response: We thank the reviewer for endorsing the publication of our work.

Reviewer #2

The authors have successfully responded to my previous concerns and should be congratulated for this very interesting study. I would suggest that you incorporate the new results with the ARB and SGLT2 inhibitor into the text because of the potential clinical relevance.

Response: We sincerely appreciate the reviewer's positive feedback. The suggested results concerning ARB and SGLT2 inhibitors have been integrated into the revised manuscript (lines 366–379).

Reviewer #3

Excellent work. All questions and concerns were resolved in my opinion.

Response: We appreciate the reviewer's positive perception of our work.